# Understanding the onset of hot streaks across artistic, cultural, and scientific careers

Lu Liu[1,2,3,4], Nima Dehmamy [1,2,3], Jillian Chown [1,3], C. Lee Giles[4,5] & Dashun Wang [1,2,3,6✉]

Across a range of creative domains, individual careers are characterized by hot streaks, which are bursts of high-impact works clustered together in close succession. Yet it remains unclear if there are any regularities underlying the beginning of hot streaks. Here, we analyze career histories of artists, film directors, and scientists, and develop deep learning and network science methods to build high-dimensional representations of their creative outputs. We find that across all three domains, individuals tend to explore diverse styles or topics before their hot streak, but become notably more focused after the hot streak begins. Crucially, hot streaks appear to be associated with neither exploration nor exploitation behavior in isolation, but a particular sequence of exploration followed by exploitation, where the transition from exploration to exploitation closely traces the onset of a hot streak. Overall, these results may have implications for identifying and nurturing talents across a wide range of creative domains.

[1] Center for Science of Science and Innovation, Northwestern University, Evanston, IL, USA. [2] Northwestern Institute on Complex Systems, Northwestern University, Evanston, IL, USA. [3] Kellogg School of Management, Northwestern University, Evanston, IL, USA. [4] College of Information Sciences and Technology, Pennsylvania State University, University Park, PA, USA. [5] Department of Computer Science and Engineering, Pennsylvania State University, University Park, PA, USA. [6] McCormick School of Engineering, Northwestern University, Evanston, IL, USA. ✉email: dashun.wang@northwestern.edu

A remarkable feature of creative careers is the existence of hot streaks[1–3]. Despite the ubiquitous nature of hot streaks across artistic, cultural, and scientific domains, it remains unclear if there are any regularities underlying the beginning of a hot streak. Understanding the origin of hot streaks is not only crucial for our quantitative understanding of patterns governing creative life cycles but it also has implications for the identification and development of talent across a wide range of settings[4,5]. Deciphering what predicts hot streaks, however, remains a challenge, partly due to the complex nature of creative careers[1,6–17]. The lack of systematic explanations for hot streaks, combined with the randomness of when they occur within a career[1], paints an unpredictable, if incomplete, view of creativity across a diverse range of domains.

Of the myriad forces that might affect career progression and success, the strategies of exploration and exploitation have attracted enduring interests from a broad set of disciplines[14–16,18–22], prompting us to examine their potential relationship with hot streaks. Indeed, according to the literature, exploitation allows individuals to build knowledge in a particular area and to refine their capabilities in that area over time. This could be relevant for understanding hot streaks since exploitation allows individuals to "go deep" in a focal area to both establish expertise in that area and foster a reputation related to that expertise[18,19]. Exploration, on the other hand, engages individuals in experimentation and search beyond their existing or prior areas of competency. Although exploration is more risky and consequently associated with larger variance in outcomes[23], it may also increase one's likelihood of stumbling upon a groundbreaking idea through unanticipated combinations of disparate sources[24]. In contrast, exploitation, as a conservative strategy, may stifle originality and, may over time, limit an individual's ability to consistently produce high-impact work[14]. Taken together, the benefits and downsides to these contrasting approaches raise a fundamental question: Are career hot streaks reflective of exploration or exploitation behavior, or some combination of the two?

To answer this question, we develop computational methods using deep learning[25,26] and network science[27,28] and apply them to large-scale datasets tracing the career outputs of artists, film directors, and scientists. Specifically, we build high-dimensional representations of the artworks, films, and scientific publications they produce (Supplementary Note 1), which capture abstract concepts, styles, and topics represented therein, allowing us to trace an individual's career trajectory on the underlying creative space (Supplementary Note 1). We further quantify the hot streak within each career by the impact of works one produced[1], measured by auction price[1,29], IMDB ratings[1,30], and paper citations in 10 years[1,12], respectively. We then correlate the timing of hot streaks with the creative trajectories for each individual, allowing us to examine changes in the characteristics of the work one produces around the beginning of a hot streak.

## Results

To examine the art styles of each artist and their exploration and exploitation dynamics, we collected over 800 K images of visual arts from museum and gallery collections, covering the career histories of 2128 artists[31,32]. Building on recent advances in computer vision[33,34], we use a transfer-learning approach[35] to construct an embedding for artworks using deep neural networks (Fig. 1a–c). We generate a 200-dimensional embedding of each artwork (see "Methods" and Supplementary Note 1.1), and identify art styles through clusters on the 200-dimensional embedding space, allowing us to trace the evolution of art styles over the course of their careers (Fig. 2a–d).

To examine the career histories of film directors, we collected our second dataset capturing plot description and cast information for each film recorded in the IMDB database (79 K films by 4337 directors; see Supplementary Note 1.2 for more detail). We build a 200-dimensional representation of each film by combining its plot and cast information (Fig. 1d, e, see "Methods," and Supplementary Note 1.2), and identify the style of each film based on clusters in the obtained embedding space, allowing us to investigate the dynamics of styles for film directors (Fig. 2e–h).

In the third setting, we analyze the career histories of 20,040 scientists by combining publication and citation datasets from the Web of Science and Google Scholar[1,12], tracing the dynamics of research topics as reflected in the publication history of each career. We use a method developed recently by Zeng et al.[16], which identifies research topics within a career by finding communities in a weighted co-citing network of all publications by the individual (Figs. 1f and 2i–l). To ensure that the results obtained for scientific careers are consistent with the embedding methods used to analyze the careers of artists and directors, we also applied a node embedding method to the co-citing network to identify research topics, and repeated our analyses, finding that the conclusions remain the same (Supplementary Note 1.3).

To quantify the exploration and exploitation behaviors reflected in each individual's career across the three domains, we measure the style or topic entropy for the work one produces, defined as $\widetilde{H} = -\sum_{i=1}^{m} p_i \log p_i$, where $p_i$ is the frequency in which one devotes to an art style or topic $i$ and $m$ is the number of unique styles or topics. On one extreme, a pure exploitation strategy means that an individual's work is contained within only one style or topic ($\widetilde{H} = 0$); on the other extreme, $\widetilde{H} = \log n$ corresponds to the case of pure exploration, where $n$ is the number of works one produced in the period, indicating that an individual's attention is evenly divided across a distribution of styles or topics ($p_i = 1/n$). For convenience, we normalize the entropy measure to obtain the rescaled entropy $H = \widetilde{H}/\log n$. Figure 2 illustrates three notable careers as examples for identifying art styles, topics, and their entropies calculated using the methodologies described above as well as in "Methods."

To test whether hot streaks are associated with exploration or exploitation, we measure the distribution of entropy $P(H)$ for works produced before and during a hot streak (Fig. 3a–c). To gauge the expected magnitude of $H$ around a hot streak, we further construct a null model for each career by randomly designating the time at which the hot streak begins[1]. We calculate the average entropy $\langle H \rangle$ measured in real careers before (Fig. 3d–f) and after the onset of the hot streak (Fig. 3g–i), and compare them with random careers, measured by the distribution of entropy, $P(\langle H \rangle)$, for 1000 realizations of the randomized careers. Figure 3d–i shows three primary findings. First, before a hot streak, $\langle H \rangle$ is systematically larger than expected (z-scores >2), indicating that individuals tend to diversify the topics they work on before a hot streak begins, consistent with an exploration strategy in the period leading up to hot streak. Second, following the onset of the hot streak, $\langle H \rangle$ measured in real careers becomes significantly smaller than expected (z-score <−2), suggesting that individuals become substantially more focused on what they work on, reflecting an exploitation strategy during hot streak. Third, despite the differences in the three types of careers we study and the methodologies to examine their career outputs, the observed associations between exploration, exploitation, and hot streaks appear universal across all three domains we studied.

To systematically examine the temporal changes in entropy, we align careers based on when their hot streak begins and measure the dynamics of $H$ around the hot streak (Fig. 3j–l). We find that compared with randomized careers, $H$ measured in real careers is

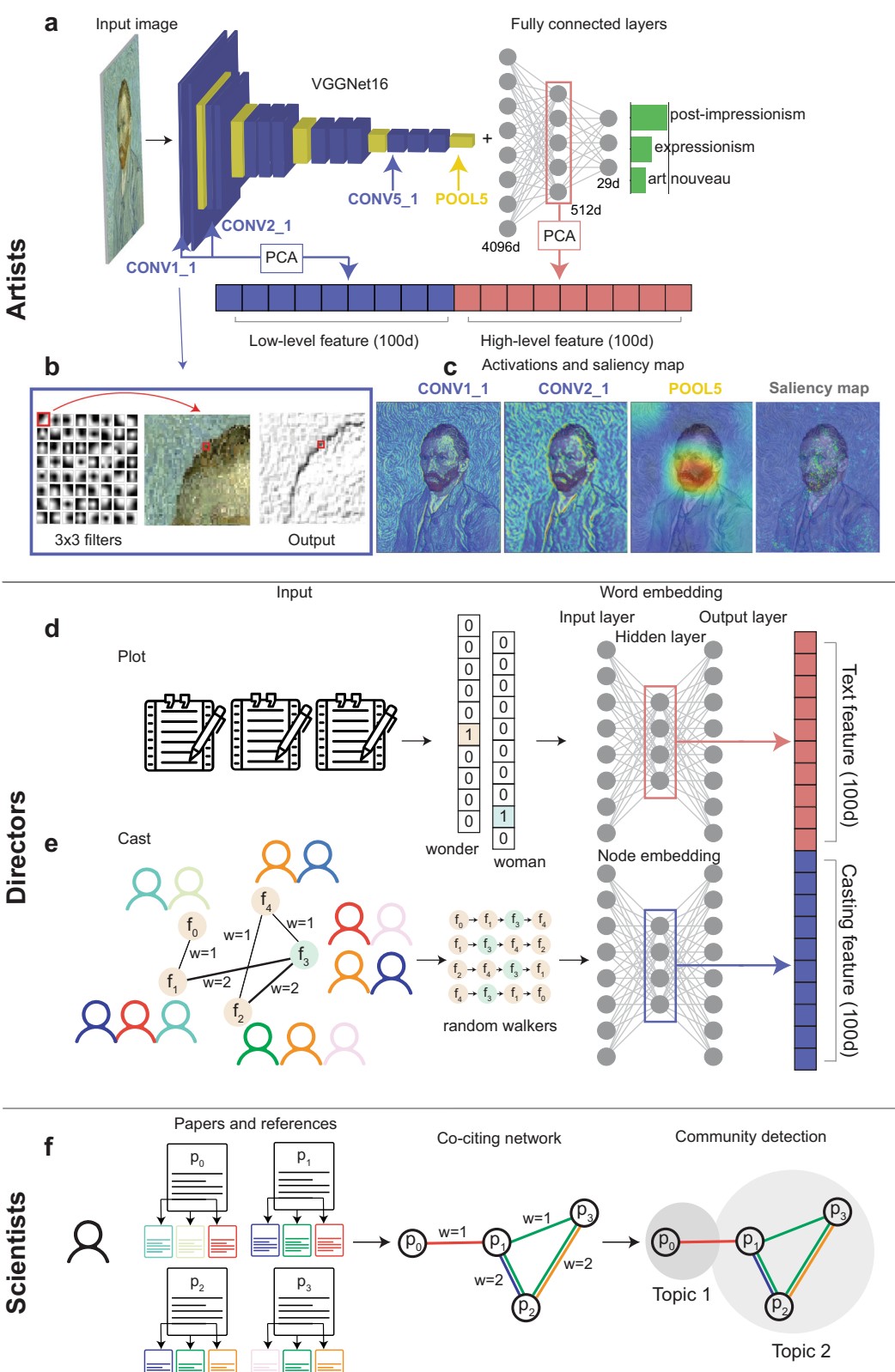

systematically elevated before a hot streak begins, but drops precipitously below expectation during the hot streak. We further compare directly the entropy distribution $P(H)$ before and after the hot streak begins, finding that, across all three domains, $H$ during a hot streak is systematically smaller than before (Fig. 3m–o, Kolmogorov–Smirnov (KS) test, $p$ value <0.001); this

pattern is absent when we repeat the same measurement for randomized careers (Fig. 3p–r).

The exploitation behavior during hot streaks appears consistent with several famous examples, including painter Jackson Pollock's "drip period" (1946–1950) (Fig. 2d), director Peter Jackson's "The Lord of the Rings trilogy" (Fig. 2h), and the career

**Fig. 1 Quantifying individual creative trajectories using high-dimensional representation techniques. a** The architecture of the deep neural network to build high-dimensional representation of artworks. We connect a pre-trained VGGNet with three fully connected layers and fine-tune the model with art style labels. The blue box indicates the convolutional layer and the yellow box the max pooling layer. The green bar shows the top styles predicted by the model for the input image (Image reproduced under Creative Commons Attribution 3.0 Unported license). We construct the high-dimensional representation of artworks by combining the output from the first and third convolutional layer (blue arrows) and the second fully connected layer (red arrow). **b** An illustration of the 64 filters in the first convolutional layer. We highlight the first filter, the original image, and the output after the image passing through the filter. The red box represents the size of the filter ($3 \times 3$ pixel box). **c** The activation of four layers in VGGNet and the saliency map of the post-impressionism class. The saliency map visualizes the important pixels for predicting the post-impressionism. Layers close to the input capture low-level features, such as brush strokes, whereas the layers close to the output capture high-level features such as the shape of objects. **d** Word embedding for film plots. Target words are encoded as a binary vector and passed to the neural network. We use the hidden layer to represent the embedding of words and plots. **e** Node embedding for the co-casting network. We apply DeepWalk to the co-casting network of 79 K films, to capture the co-occurrence of nodes from the trajectories of random walkers. We use the hidden layer of the model to represent the cast information. We concatenate the word embedding from plots and the node embedding from casts to construct a 200-dimensional vector to represent each film. **f** An illustration of the co-citing network among papers published by a scientist. Two papers are connected if they have at least one common reference, with link weight measuring the total number of references they share. Following prior work[16], we apply a community detection algorithm to the co-citing network and identify the topic of each paper as the community it belongs to.

of scientist John Fenn, whose hot streak arrived late in his career, but the work he produced during that period on electrospray ionization eventually won him the chemistry Nobel in 2002 (Fig. 2l). These examples raise an intriguing question: can the exploitation behavior by itself predict career hot streaks? To test this, we identify episodes of exploitation in each career by tracing the dynamics of $H$ across our three domains. We calculate the probability of initiating a hot streak with the onset of an exploitation episode, and compare it with the baseline probability measured in randomized careers (Fig. 3s–u). We find that when exploitation occurs by itself, not preceded by exploration, the chance that such episodes coincide with a hot streak is significantly lower than expected, not higher, across all three domains. These results indicate that exploitation by itself may not guarantee hot streaks, further suggesting the importance of prior exploration. Indeed, reexaminations of the careers of Jackson Pollock, Peter Jackson, and John Fenn reveal a phase of unusual exploration of new and diverse art styles, types of films, and research topics, respectively, for the period leading up to their hot streaks (Fig. 2c, g, k). This observation raises the question of whether exploration that precedes a hot streak is instead the crucial ingredient, prompting us to calculate the probability of initiating a hot streak following an exploration episode alone. However, we find that when the episode of exploration is not followed by exploitation, the chance for such exploration to coincide with a hot streak again reduces significantly. By contrast, exploration followed by exploitation appears consistently associated with a significant lift in the probability of initiating a hot streak: this configuration consistently outperforms the baseline across all three domains (20.5%, 13.8%, and 19.2% over the baseline for artists, directors, and scientists, respectively), and represents the only positive lift among all combinations of the two creative strategies (Fig. 3s–u). Figure S46 further examines the exploration, exploitation, and normal phases, and explores all potential sequences of any two of the three phases (nine in total), reaching the same conclusions.

Taken together, these results suggest that neither exploration nor exploitation alone is associated with the hot streak dynamics; rather, it is the shift from exploration to exploitation that closely traces the onset of a hot streak. One plausible explanation is that exploration, as a risky, variance-enhancing strategy, increases one's chances to stumble upon new, potentially groundbreaking ideas; the subsequent exploitation behavior allows the individual to focus, develop knowledge and capabilities in that focal area, and build out their discoveries further. Importantly, our findings suggest that both ingredients of exploration and exploitation seem necessary. This supports the notion that not all explorations are fruitful, and that

exploitation in the absence of promising new ideas may not be as productive. On the other hand, the sequence of exploration followed by exploitation may facilitate the emergence of high-impact work by incorporating new insights into a focused agenda. The positioning of exploration before exploitation may therefore serve to expand an individual's creative possibilities.

We test the robustness of our results across several dimensions. We split our samples of artists, directors, and scientists based on the timing of their hot streaks (Supplementary Note 3.1), the individual's level of impact (Supplementary Note 3.2), and different fields of studies (Supplementary Note 3.3), and repeat our analyses in each subsample, arriving at consistent conclusions. We further control for individual fixed effects in their exploration–exploitation dynamics (Supplementary Note 3.4), and find that artists, directors, and scientists predictably deviate from their typical creative behaviors around the beginning of a hot streak: individuals who tend to exploit become more exploratory before a hot streak begins, whereas individuals who tend to explore become particularly focused during their hot streak (Supplementary Note 3.4). We further use regression analysis to fit the relationship between hot streaks and the exploration–exploitation transition by controlling for the impact of an individual's work, their career stage, and other individual characteristics, and find that our conclusions remain the same (Supplementary Note 3.5). For scientists who experience two hot streaks, we perform our measurements for the first and second hot streak separately (Supplementary Note 3.6), and find that the exploration–exploitation dynamics hold true in both cases. For those having hot streaks at the beginning of their careers, while by construction we cannot observe their prior behaviors, we find that they consistently engage in exploitation during their hot streaks (Supplementary Note 3.7). We further verify that these results are robust to using different community detection algorithms such as Infomap[36] (Supplementary Note 3.8) and different ways of aggregating data over time (Supplementary Note 3.9). We also replaced our entropy measure to quantify the exploration–exploitation dynamics by the Simpson diversity measure $(1 - \Sigma_i p_i^2)$ (Supplementary Note 3.10), the number of styles or topics (Supplementary Note 3.11), the fraction of works in the most popular style or topic (Supplementary Note 3.12), and probability of switching topics (Supplementary Note 3.13), and repeat all our analyses, finding again the same conclusions.

To understand the potential forces that might facilitate the shift from exploration to exploitation, we further examine the organization of innovative activity. Motivated by the literature on science teams[8,37,38], here we focus on scientific careers only, asking whether there are detectable changes in collaboration patterns around the exploration–exploitation transition. We find

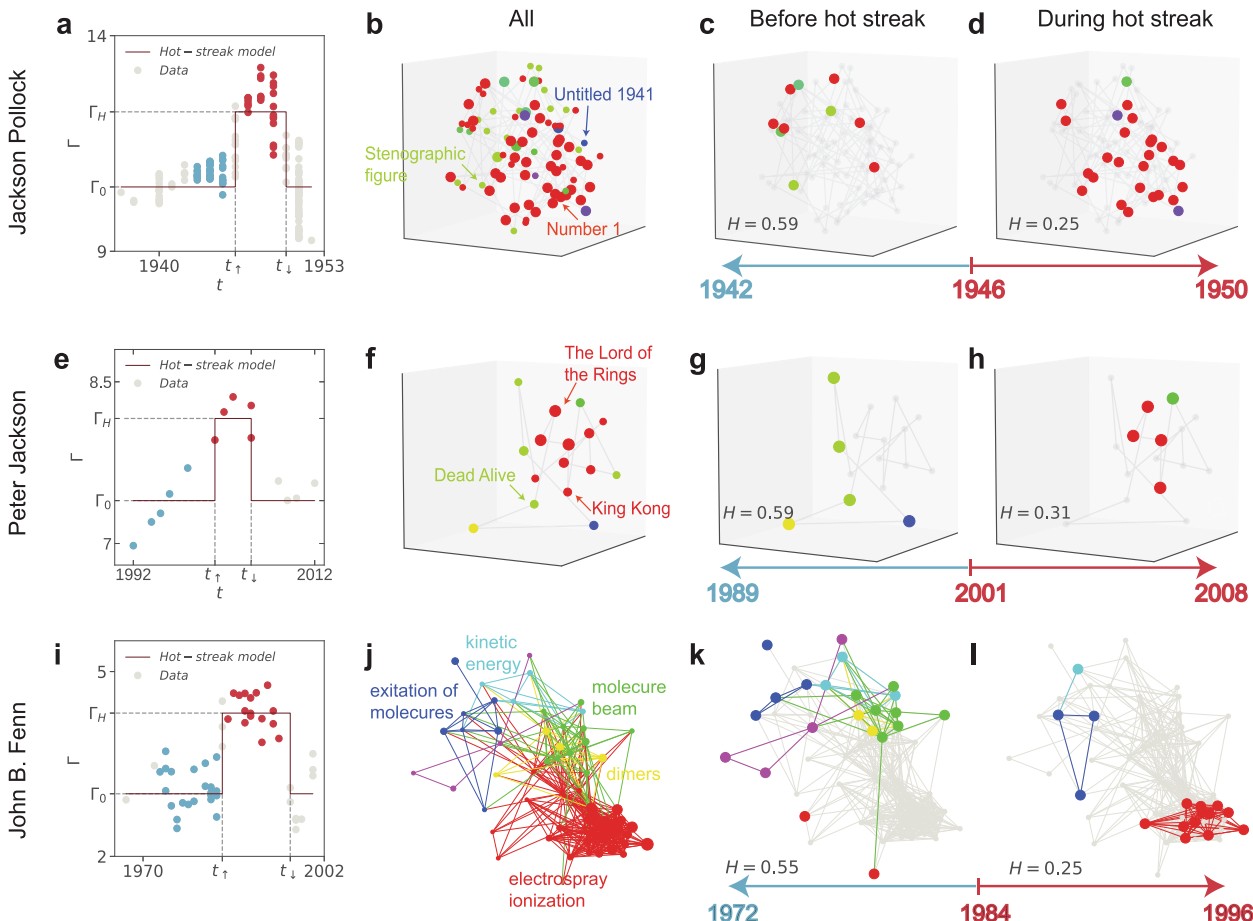

**Fig. 2 Creative trajectories and hot-streak dynamics: three exemplary careers. a, e, i** We fit the hot-streak model to careers of **a** Jackson Pollock, **e** Peter Jackson and **i** John B. Fenn. The hot-streak model[1] assumes that the impact of works produced in a career (log price for artworks, IMDB rating for films and $\log C_{10}$ for papers) is drawn from two normal distributions ($\Gamma_0$ and $\Gamma_H$), where $\Gamma_0$ captures the typical performance and $\Gamma_H$ captures the performance during hot streak. The red line denotes the hot-streak model. $t_\uparrow$ and $t_\downarrow$ marks the beginning and the end of hot streak. To avoid mixing across the two periods, we measure the entropy of styles or topics for works produced before and during hot streak by excluding those produced during the year of the transition. **b–d** We project the 200-dimensional representation of artworks produced by Jackson Pollock to a 3D t-SNE embedding space. Different styles are shown in different colors, and nodes with larger sizes denote those produced during hot streak. For Jackson Pollock, his hot streak is well aligned with the famous "drip period" (1946–1950). The entropy of works produced during this period is substantially lower than typical ($H = 0.25$ vs $H = 0.43$), suggesting an intensive focus on one particular style (**d**). This exploitation behavior contrasts the work he produced in the period leading up to hot streak, which was characterized by an unusual exploration of new and diverse styles ($H = 0.59$) (**c**). **f–h** We project the films vectors produced by Peter Jackson to a t-SNE embedding space. Peter Jackson's hot streak covers "The Lord of the Rings" trilogy ($H = 0.31$) (**h**). Before his hot streak, however, Jackson worked on diverse types of films including biography and horror-comedy ($H = 0.59$) (**g**). **j–l** For the career of John Fenn, we study the co-citing network of his papers. Before his hot streak, Fenn worked on numerous different topics from excitation on hot surfaces to dimers ($H = 0.55$) (**k**). But during his hot streak, Fenn intensively focused on electrospray ionization ($H = 0.25$) (**l**), which eventually won him the chemistry Nobel in 2002.

that scientists are more likely to explore with small teams before a hot streak, but exploit with large teams after a hot streak begins. Indeed, we quantify the change in team size through two measures. We trace the dynamics of team size around the beginning of a hot streak (Fig. 4a). We also calculate the team size distribution observed in real careers normalized by the randomized careers ($R$(team size); Fig. 4b). Both results show that team size drops significantly before the hot streak yet becomes substantially larger than expected during the hot streak (Fig. 4a, b). We further find that the onset of hot streaks appears to mark an increase in new collaborators (Supplementary Note 4), consistent with the advantages of fresh teams[38]. Note that the role and definition of teams vary substantially across the three domains, hence this analysis is applicable to scientific careers only. Given the observational nature of our study, we cannot rule out potential omitted variables that might mediate these patterns. Nevertheless, these results are in line with the findings that small and large teams are

differentially positioned for innovation[37]: large teams tend to excel at furthering existing ideas and design, whereas small teams tend to disrupt current ways of thinking with new ideas and opportunities. We further test the robustness of these results across different disciplines, adjusting for self-citations, and controlling for the publication year, research field, and career stage using regression analysis, all arriving at the same conclusions (Supplementary Note 4).

Our next analysis probes potential connections between phases of exploration and exploitation surrounding a scientist's hot streak. We examine properties of the topics that are explored during the period leading up to hot streak, ranging from recency to citation impact to popularity, asking which topics tend to be chosen for subsequent exploitation. We find that the topic that was eventually exploited is less likely to be the one explored the most recently, or the highest cited, or the most popular among the topics explored before (see Supplementary Note 5). These

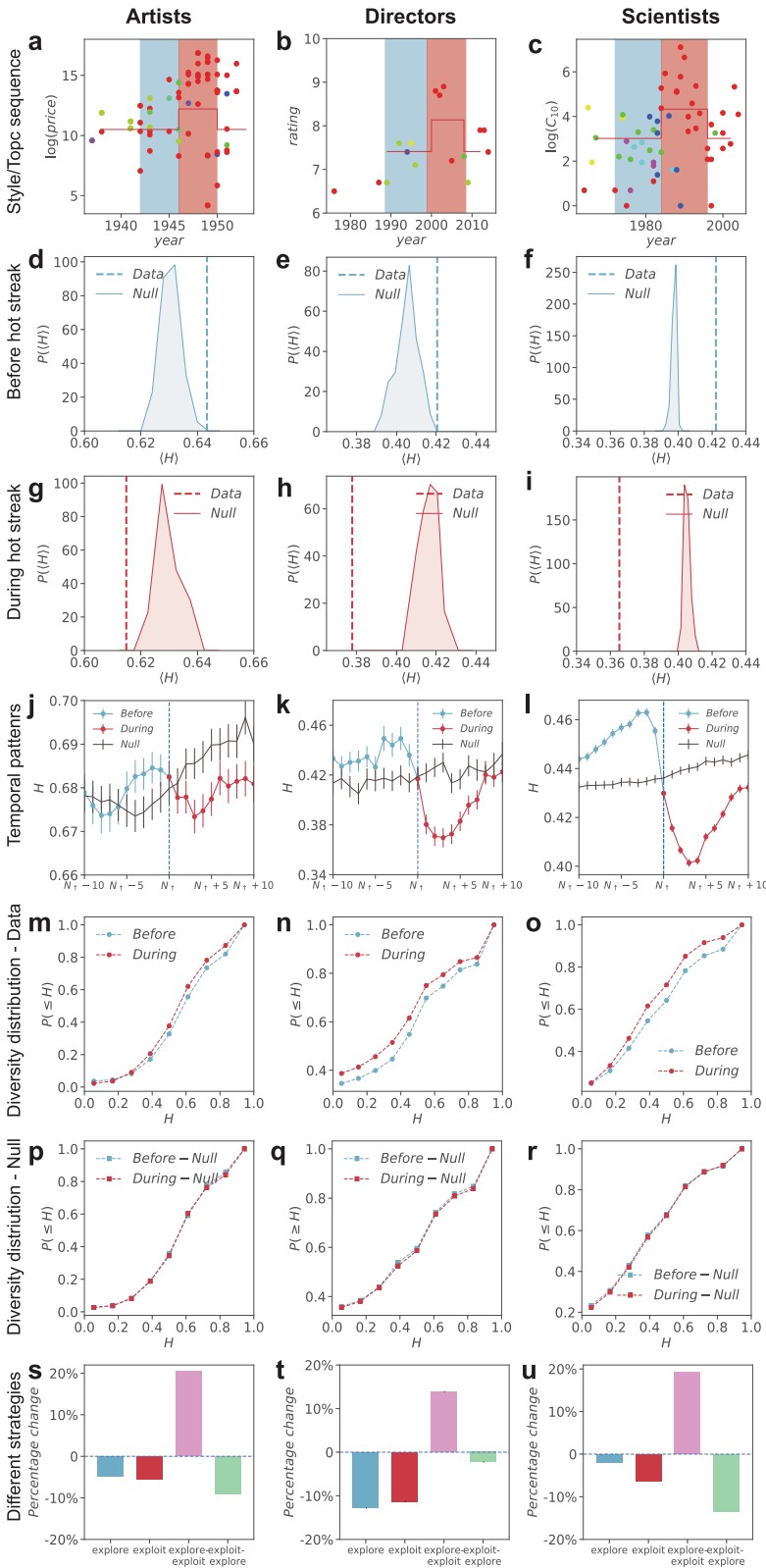

findings imply that, more than simply chasing after discovery through exploration, individuals appear to seek out new opportunities by deliberating over different possibilities, and then harvesting promising directions through exploitation. To test if these potential connections can help us better understand which direction to exploit following exploration, we set up a simple prediction task to predict which topic to exploit using the features discussed above that characterize the exploration phase, including team size and topic properties (Supplementary Note 5); this exercise yielded substantial predictive power (accuracy of 0.89 and area under the curve of 0.83). Overall, these results suggest intriguing connections between phases of exploration and

**Fig. 3 Exploration, exploitation and career hot streaks. a–c** Career histories of **a** Jackson Pollock, **b** Peter Jackson, and **c** John Fenn illustrate the topics they worked on before and during their hot streak and the impacts of the work. Color of the dots is consistent with the dots shown in Fig. 2b, f, j. **d–f** The distribution of entropy $P(\langle H \rangle)$ before a hot streak for 1000 realizations of the randomized careers for all individuals analyzed in our datasets. The vertical line indicates $\langle H \rangle$ measured in real careers, showing that it is significantly larger than expected ($z$-scores are 4.24 for artists, 2.94 for directors, and 13.90 for scientists). **g–i** Same as (**d–f**), but for the entropy of work produced during hot streak. $\langle H \rangle$ in real careers (vertical line) is significantly smaller than expected ($z$-scores are −2.42 for artists, −8.54 for directors, and −22.71 for scientists). **j–l** The dynamics of topic entropy $H$ surrounding the onset of hot streak for real and randomized careers, measured through a sliding window of six artworks, five films or five scientific papers. Error bars represent the standard error of the mean. **m–o** Cumulative entropy distribution $P_{\leq}(H)$ before and during hot streak in real careers across the three domains. $P$ values of the KS-test are $3.7 \times 10^{-6}$ for artists, $1.5 \times 10^{-5}$ for directors, and $1.1 \times 10^{-64}$ for scientists. **p–r** Cumulative entropy distribution $P_{\leq}(H)$ before and during hot streak for the null model. $P$-values are 0.23 for artists, 0.77 for directors, and 0.06 for scientists. **s–u** The probability to observe the onset of a hot streak at the end of an exploration episode alone (not followed by exploitation), or at the beginning of an exploitation episode alone (not proceeded by exploration), or at the transition from exploration to exploitation, or from exploitation to exploration. We then compare with the baseline probability of having a hot streak. Here we calculate entropy with a sliding window of two years for artists and scientists, and five works for directors, and define exploration and exploitation episodes as entropy above or below one's average.

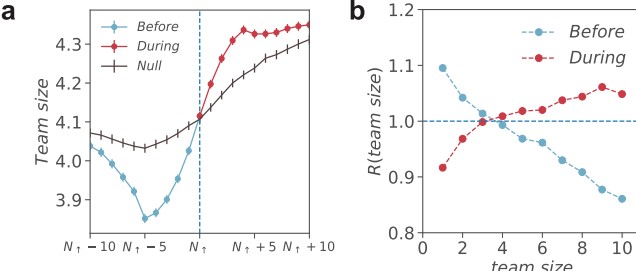

**Fig. 4 Authorship structure and hot streaks in science. a** The average team size around the beginning of a hot streak for real and randomized careers in science. Team size shows a significant drop before a hot streak begins, but a notable increase during hot streak. **b** We calculate team size for papers published before and during hot streak, and compare the distribution to that of randomized careers for 500 realizations, denoted as $R$(team size). $R$ decreases with team size from above to below 1 for papers published before a hot streak but increases with team size for papers published during a hot streak. Both measures in (**a**) and (**b**) suggest that scientists tend to engage with smaller teams before hot streak, and with larger teams during the hot streak. Error bars represent the standard error of the mean.

exploitation surrounding a hot streak, which may have implications for science funding, especially given hot streaks and research grants tend to last for a similar duration.

Finally, we consider career trajectories following the end of a hot streak. We measure the average entropy $\langle H \rangle$ after the end of a hot streak and compare the measurements in real careers with the distribution of entropy $P(\langle H \rangle)$ from the randomized careers (Fig. 5a–c). We find that, after the hot-streak period, $\langle H \rangle$ becomes statistically indistinguishable from the randomized careers ($-1 \leq z$-score $\leq 1$). We further examine the temporal changes in entropy at the end of a hot streak by aligning careers based on when their hot streaks end (Fig. 5d–f). We find again a lack of difference between data and the null model. Together, these analyses suggest that individuals return to "normal" after their hot streak ends, showing an absence of exploration or exploitation patterns.

## Discussions

Taken together, these results unveil identifiable regularities underlying the onset of career hot streaks, which appear to apply universally across a wide range of creative domains. Overall, our results highlight the important role of both exploration and exploitation in individual careers. Curiously, across all three domains we studied, a major turning point for individual careers appears most closely linked with neither exploration nor

exploitation behavior in isolation, but rather with the particular sequence of exploration followed by exploitation. Indeed, extant literature has documented the fundamental role of exploration and exploitation in creativity (Supplementary Note 2.2 and Supplementary Table 1). Yet as creative behaviors, they have traditionally been considered either in isolation or in combination but rarely in succession[14,22]; this is especially the case for career-level analysis. Our results suggest a sequential view of creative strategies that balance experimentation and implementation may be particularly powerful for producing long-lasting contributions. These findings may hold broad relevance for identifying, training, and nurturing creative talents, especially given the various forces that sometimes appear in tension with the exploration–exploitation dynamics, ranging from the intensifying pressure to publish[39,40] to the increasing trend of exploration over a career[16], from the specialization of individual expertise[10] to how such specialization is favored in personnel evaluations[41,42].

It is important to note that while our results demonstrate significant and consistent relationships across domains, the overall effect size seems modest. On the one hand, this suggests that additional controls might further tighten the relationship. For example, after we control for authorship and the effect of collaborations, the effect size seems to magnify (Supplementary Note 4.5). On the other hand, it also suggests opportunities to examine other potential processes that may also underlie the onset of hot streaks. Indeed, real careers are complex, with heterogeneous influences operating across domains as well as a multitude of individual and institutional factors. Hence, it is plausible that additional factors may also be at work. In this study, we also tested several alternative explanations for the onset of hot streaks (Supplementary Note 6). Although each of these hypotheses we tested appears plausible by itself, we find that none of them shows consistent associations, indicating that none of these alternative hypotheses alone can account for the hot-streak dynamics we studied. It is also likely that on an individual basis, the exploration–exploitation transition is further influenced by other external factors, such as shifting market conditions[43], social network structure[38,44], and disciplinary culture[18,19]. Individuals may also receive short-term feedback (e.g., art critiques or peer reviews) that may offer additional signals shaping their career focus. As such, the patterns of exploration and exploitation may reflect personal initiatives as well as responses to external forces. Nevertheless, our results suggest that, despite the obvious heterogeneity in the settings we examined and the myriad factors that may affect career progression and success, the exploration–exploitation dynamics appears consistently associated with the onset of hot streaks across rather diverse domains.

The data-driven nature of our study indicates that it is not immune to two limitations common in this type of analysis. First,

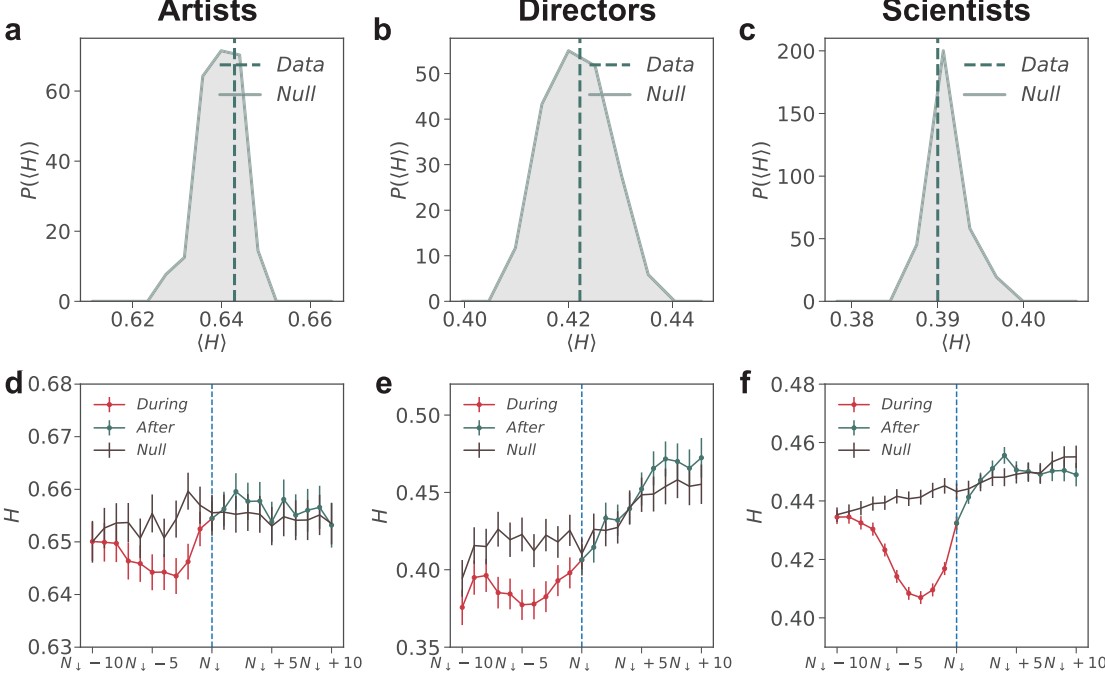

**Fig. 5 When hot streak ends. a–c** The distribution of entropy $P(\langle H \rangle)$ after a hot streak for 1000 realizations of the randomized careers for all individuals analyzed in our datasets. The vertical line indicates $\langle H \rangle$ measured in real careers, showing that it is statistically indistinguishable from the randomized careers. **d–f** The dynamics of entropy $H$ surrounding the end of hot streaks for real and randomized careers, measured through a sliding window of six artworks, five films, or five scientific papers. Error bars represent the standard error of the mean.

while the datasets we assembled in this paper represent large collections of career histories and outputs across a variety of domains, they are limited to individuals who have had sufficiently long careers providing enough data points for statistical analyses (Supplementary Note 1). Second, this paper presents correlational evidence, whose primary goal is to investigate empirical regularities associated with the onset of hot streaks. Future work using causal research designs may improve causative interpretations of the regularities reported here.

Furthermore, while this work mainly focuses on universal patterns related to the onset of hot streaks, there could be important domain-specific differences in the role of exploration, exploitation, and success that are worth investigating further. For example, our preliminary analysis suggests that the level of exploration and exploitation in science appears much stronger than in art or film directing (Fig. 3). The number of styles/topics within each career also varies substantially across domains (Supplementary Fig. 30). While these cross-domain differences could flow from inherent differences in data and methods, assessing domain-specific patterns is an important direction for future work.

Notably, the sequence of exploration followed by exploitation closely resembles strategies observed in a wide range of natural and socio-technical settings, from animal foraging[45] to human cognitive search[46], from multi-armed bandits and reinforcement learning[47] to role oscillation between brokerage and closure in social network[48] to changing innovation strategies over business cycles[49]. It thus suggests that the sequential strategies of exploration followed by exploitation uncovered in this study may have broad relevance that goes beyond individuals' careers. Lastly, the representation techniques used in this paper could open up promising avenues for research on creativity[50–52], offering a quantitative framework to probe the characteristics of the creative products themselves. Future advances in deep learning may

enable researchers to incorporate more creative dimensions, and hence more fruitfully contribute to a computationally enhanced understanding of creativity.

## Methods

**High-dimensional representation of artworks**. We apply a pre-trained VGGNet algorithm[33], one of the best-known algorithms for image recognition, to images of artworks, and connect it with an additional neural network with fully connected layers to classify the art style labels recorded in our dataset (Fig. 1a). The convolutional layers in the pre-trained VGGNet use 3 × 3 filters to detect local patterns from the artwork (Fig. 1b). The filters in the first layer capture spatial patterns such as line orientations and brushstrokes (Fig. 1b), whereas those in higher layers combine outputs of filters from lower layers to capture more complex features, such as shapes and objects (Fig. 1c). To leverage VGGNet's image recognition capabilities, here we do not train the VGGNet layers, but instead train the fully connected layers to repurpose VGGNet to identify art styles (Fig. 1a), helping the first two fully connected layers to find an abstract representation of concepts and themes by grouping together related outputs of the VGGNet layers. Prior research shows that art style may be decoded from both brush strokes and the overall concepts, subjects, and themes[34,53], suggesting that both low- and high-level features are important for capturing art styles. We combine the outputs from the first and third convolutional layers in VGGNet with the fully connected layer before the final classification layer (see Supplementary Note 1.1 for several case studies showing how art styles are interpreted by our deep learning framework). We apply our deep neural network to the career outputs of each artist in the dataset, and then use principal component analysis for dimensionality reduction to generate a 200-dimensional embedding of each artwork.

**High-dimensional representation of films**. We build high-dimensional representations of films by combining the plot and casting information of each film. We first train word embeddings[51] in the description of the plot to learn a 100-dimensional text representation of a film from the co-occurrence of words (Fig. 1d and Supplementary Note 1.2). To incorporate casting information, we construct a weighted co-casting network among all actors and apply a node embedding method DeepWalk[54] to obtain a 100-dimensional casting vector for each film (Fig. 1e and Supplementary Note 1.2). We then concatenate the vectors for plot and cast, allowing us to develop a 200-dimensional embedding space to represent all films. Despite the myriad factors that may affect the artistic and financial success of a film[55], ranging from the screenplay to acting, we find that the learned high-

dimensional representation can successfully predict film genre with an accuracy of 0.948 (Supplementary Note 1.2).

**Reporting summary**. Further information on research design is available in the Nature Research Reporting Summary linked to this article.

## Data availability

The data used in this study have been deposited in the GitHub repository https://kellogg-cssi.github.io/onsethotstreaks.

## Code availability

The code used in this study has been deposited in the GitHub repository https://kellogg-cssi.github.io/onsethotstreaks.

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

## Acknowledgements

We thank A.-L. Barabási, W. Ocasio, B. Uzzi, J. Evans, K. Rao, C. Candia, S. Medya, G. Tripodi, and all members of the Center for Science of Science and Innovation (CSSI) for invaluable comments. This work is supported by the Air Force Office of Scientific Research under award numbers FA9550-15-1-0162, FA9550-17-1-0089, and FA9550-19-1-0354.

## Author contributions

D.W. conceived the project and designed the experiments; L.L. and N.D. collected data and performed empirical analyses with help from J.C., C.L.G. and D.W.; all authors discussed and interpreted results; D.W., L.L. and N.D. wrote the manuscript; all authors edited the manuscript.

## Competing interests

The authors declare no competing interests.
