## [Peer Review File · Nature Communications]

Reviewers' Comments:

Reviewer #2:

Remarks to the Author:

This paper presents a systematic study on the characteristics change before and during hot streaks across artistic, cultural, and scientific careers. This topic is unique and interesting to career development, and has high impact on various fields like academia, business, marketing, and artists. Although it may not be surprising that exploration followed by exploitation is a key evolution of hot streaks, the main contribution of this work is the large-scale and systematic study with statistical analysis from different perspectives. I think this work is original, and presents a methodology for studying exploration and exploitation in career development.

The main content is well written, and the supplemental materials provide rich and various information to show the claims. The authors provide codes, and thus researchers should be able to reproduce the work. The datasets used in this paper can also be collected by other researchers.

I am not really in the field of this type of study, but am more involved with visual analysis. The method used for artistic style is quite standard and convincing.

One minor comment on the main content is the visual quality of figures. Higher resolution of images can be provided to avoid some blur effects.

Reviewer #3:

Remarks to the Author:

This paper presents an interesting attempt to study the causes of "hot streaks" in success in three domains -- art, science, and movie direction. They demonstrate that hot streaks generally are periods of focused "exploitation" following less focused "exploration".

The study uses some clever technical tricks (like image/graph embeddings to measure similarity) over diverse domains and the conclusions generally make sense.

There are natural questions of cause and effect, however, which vary by domains. Certain observations about team size fall in this regard. If I am successful at research, I get more grants - thus I can build bigger teams. If I make a successful movie, I will be given the chance to make another successful movie like it until my movies lose money. Auction prices of paintings typically reflect sales many years after the work has been made. If one painting sells for a lot, similar paintings of the same period should also sell well by definition.

I cannot help but wonder if the effects seen here are basically just "if something is successful, do it again". Peter Jackson makes a second Hobbit movie because the first is successful. If one of my paintings sell, make another just like it and see if it sells. If I am happy with the results of a paper, keep working on it. That would suggest an autoregressive model would generally predict success from the previous works.

One important thing left undone is a study of what happens after hot streaks. Do they lead to a resumption of exploration, or is that people stay in the domain trying to hit lightning again? I would encourage the author to report the analysis in the next revision.

Reviewer #4:

Remarks to the Author:

The paper by Liu et al. studies hot streaks in various types of careers and attempt to understand the patterns that occur prior to the onset of a hot streak. The authors compare the exploration and exploitation of an individual before and during a hot streak, finding that before the hot streak individuals tend to carry out more exploration and during a hot streak they have increased exploitation.

The results are robust across numerous statistical tests and controls, that the authors carry out. Overall, I believe that the results are very interesting, and the manuscript can be published after the authors consider the following comments.

1. While the results are consistent and significant. The effect size of exploration prior to exploitation is not particularly large in many cases. For example, the increase in the likelihood of a hot streak is 10-20% given that one follows the pattern of exploration-exploitation. Similarly, many of the comparisons of the distribution of entropy $\langle H \rangle$ compared to the null model show significant but not necessarily dramatic differences (see e.g. Fig. 3d-3i and Figs. S25-S32). This would suggest that individuals somewhat pare down their focuses during the hot streak, but not dramatically so. Is it possible that within the period of the hot streak there is some control that could be done to highlight the exploitation of one particular area where the individual is having the greatest success yet still continuing, perhaps in a lesser role, in other areas? For scientific careers, this could be assessed by authorship order where perhaps the increased impact occurs in a single area where the researcher is acting as a senior author while still continuing collaborations in other areas? Alternatively, is the increased performance during the hot streak occurring in multiple focus areas simultaneously?
2. While entropy is a natural metric for measuring the diversity of topics and the level of focus, it is not always easy to interpret. Would it be possible to also add some simpler measures to assess the variety of topics before and during the hot streak? For example, could the authors show what fraction of an author's works are in the most popular topic and what is the distribution of how many topics an author works on before and during the hot streak? These could also be compared to the null model and could help readers' comprehension beyond the entropy metric.
3. Regarding the sizes of teams, the authors only explored this area for scientists. I realize that artists work alone and thus there is assessing team size there, however, is it possible to assess team size for film directors since films typically result from many individuals? Even if films are more fixed in their overall team size due to the many parts involved, it would at least be interesting to show a graph similar to Fig. S36 for films. In this context also seem relevant the recent study of An Zeng et al arXiv:2007.05985 which shows the effect of new collaborators.
4. Regarding the team sizes, can the authors add error bars to the graphs in Fig. 4, or how large is the deviation in team sizes on the various points?
5. While the hot streak model is based on prior work, can the authors justify in the SI the use of two normal distributions for representing the typical performance and elevated performance? Generally, performance in these types of domains tends to have a long tail (power-law), and thus would it not make more sense to consider for example two power-law distributions?

Other minor points:

1. On page 5 in the paragraph beginning "To systematically examine..." the phrase "during hot streak" should be "during a hot streak".
2. In the caption to Fig. 1 in the last sentence, "we apply community detection algorithm" should be "we apply a community detection algorithm".
3. In Fig. 3 is it possible to either separate the x-axis labels more? Currently, it is difficult to understand how to read the labels that are on two lines.

Reviewer #5:

Remarks to the Author:

The main contribution of this work is to apply advanced computational methods of deep learning and network science to build high-dimensional representations of creative works grouped into career trajectories, thereby affording analysis of the transition into a hot streak. On this merit alone, the manuscript deserves consideration. However, upon a close read, many conclusions are presently based upon the argument that consistent results across the three domains somehow is justification in of itself. It would be more satisfying to see results that differentiate between the domains that are consistent with fundamental differences in the role of exploration, exploitation and success in each.

Detailed comments:

The authors focus on March's exploitation-exploration dichotomy to resolve some of the open questions identified in their parent paper "Hot streaks in artistic, cultural, and scientific careers" (Liu et al, Nature 2018), where they reported that "The hot streak emerges randomly within an individual's sequence of works, is temporally localized, and is not associated with any detectable change in productivity." Here they elaborate on this prior work by analyzing shifts in creative exploitation-exploration strategy occurring at the onset of hot streaks. The authors rely on statistical comparison with randomized data to arrive at most of their conclusions. Yet there are more sophisticated means of identifying causal effects, and so it's not clear why they resort to distribution-level comparisons with shuffled data and the like. As elaborated below, it's also not clear how the authors account for biased estimates arising from highly correlated observations.

One of the main results highlighted in the abstract is "a particular sequence of exploration followed by exploitation, where the transition from exploration to exploitation closely traces the onset of a hot streak". While at first this statement may cause pause for reflection, it's not clear in what scenario one would expect the reverse, exploitation followed by exploration, to be a reliable alternative strategy. If exploration sets the conditions for arriving at novel creations, then it seems that a better summary of the results is that the onset of hot streaks are marked by more novel creations, which again is not surprising. It appears as though the hot streak is more of a doubling-down on intermittent success more than randomly arriving at King Midas' golden touch.

This consideration brings forth a bigger question of survival bias regarding the many explorers who were ejected because they did not identify the 'right' novel creation before their time to capitalize on the creation evaporated. This point should also be considered regarding the selection of visual arts analyzed from museum and art gallery collections, which depend on strict economically-motivated selection criteria. What about the art works that were not shown? Would one expect them exhibit the same exploration-exploitation patterns as the exhibited work? This would provide more convincing evidence that the source of the strategy shift has its origins with the creator. Recent research by Baliotti et al. (Peer review and competition in the Art Exhibition Game, PNAS 2016) shows how competitive exhibition can alter the perceived evaluation of creative works. So do creators transition into the more lucrative exploration state as a result of their own inclination or in response to the external market? Or in terms of the author's statement "On the other hand, the sequence of exploration followed by exploitation may facilitate the emergence of high-impact work by incorporating new insights into a focused agenda", is the agenda driven by the artist or the artgoers? Or in the context of the collaboration sizes, is the hot streak driven by the scientist or her new collaborators? In this regard, the analysis has the effect of raising more questions than it answers.

In summary, this work provides an almost overwhelming body of evidence that is counter to what was originally reported, that hot streaks are not associated with any detectable change in productivity; the authors are now repositioning hot streaks as being clearly associated with changes in productive strategy as they relate to team size. These new results would indicate that the hot streak has more to do with the market conditions for success than the individual creator.

Additional comments:

A) It's not clear why exploitation would consist of in film directing that would be measurable given the data collected. The motivation and definition of what exploitation consists of in each profession is not clear. The authors seem to take it as a given, and so the rhetorical statement "Are career hot streaks reflective of exploration or exploitation behavior, or some combination of the two?" seems self-fulfilling.

B) The authors use entropy measures to quantify exploration versus exploitation. This is an odd choice, albeit convenient, as it measures diversity which does not necessarily distinguish exploration from exploitation. An individual may explore a single domain by going deeper. A metric that captures transitions between topics would better capture the phenomena being analyzed. Another issue is that the number of topics m is a variable that may vary between domains, but there is little discussion of how this impacts the results. Fig S14, 17 and 18 showing the

distributions $P(H)$ indicate that the location of H values are highly variable. On closer inspection, the difference between the location of the null line and the bulk of the $P(H)$ distributions are relatively small also.

C) Are the authors measuring hot streaks in which creators have the King Midas touch, or are they just capitalizing on variations around a successful theme? Peter Jackson's career exhibited in Fig 2 provides a good example, where it appears as though the Lord of the Rings episodes are treated as 3 independent observations? The same could be argued for all paintings belonging to Pollock's drip period - are these to be considered separate or variations on the same theme? How much do these types of correlated observations explain the main results of the analysis?

D) Given the authors' definitions of exploration and exploitation based upon extreme z-scores, the statements "We find that when exploitation occurs by itself" and "following an exploration episode alone" are unclear, since an individual cannot be in a hybrid exploration and exploitation state.

E) The analysis regarding team size suffers from endogeneity issues commonly attributable to multiple explanatory variables - does the transition into the exploration period produce more diverse work that is more highly cited for this reason, or does the shift to larger teams produce more diverse work that is more highly cited for this different reason, not least of which is there are more coauthors to show the work.

Point-By-Point Response

Reviewer #2:

2.1 This paper presents a systematic study on the characteristics change before and during hot streaks across artistic, cultural, and scientific careers. This topic is unique and interesting to career development, and has high impact on various fields like academia, business, marketing, and artists. Although it may not be surprising that exploration followed by exploitation is a key evolution of hot streaks, the main contribution of this work is the large-scale and systematic study with statistical analysis from different perspectives. I think this work is original, and presents a methodology for studying exploration and exploitation in career development.

The main content is well written, and the supplemental materials provide rich and various information to show the claims. The authors provide codes, and thus researchers should be able to reproduce the work. The datasets used in this paper can also be collected by other researchers.

Response: We wish to thank the Reviewer for his/her positive assessment of the manuscript, as well as for highlighting the novelty and technical soundness of our approach. We are grateful for and encouraged by the enthusiastic endorsement by the Reviewer.

2.2 I am not really in the field of this type of study, but am more involved with visual analysis. The method used for artistic style is quite standard and convincing.

Response: We thank the Reviewer for recognizing the method used for artistic styles as standard and convincing, which further assures us of the technical soundness of the paper.

2.3 One minor comment on the main content is the visual quality of figures. Higher resolution of images can be provided to avoid some blur effects.

Response: We thank the Reviewer for pointing out this issue. We have updated all the figures with higher resolution in the revised manuscript.

Reviewer #3

3.1 *This paper presents an interesting attempt to study the causes of "hot streaks" in success in three domains -- art, science, and movie direction. They demonstrate that hot streaks generally are periods of focused "exploitation" following less focused "exploration".*

The study uses some clever technical tricks (like image/graph embeddings to measure similarity) over diverse domains and the conclusions generally make sense.

Response: We wish to thank the Reviewer for the overall positive assessment of the paper and for appreciating the techniques we used across diverse domains. Next, we offer a detailed point-by-point response to your insightful questions and suggestions.

3.2 *There are natural questions of cause and effect, however, which vary by domains. Certain observations about team size fall in this regard. If I am successful at research, I get more grants -- thus I can build bigger teams. If I make a successful movie, I will be given the chance to make another successful movie like it until my movies lose money. Auction prices of paintings typically reflect sales many years after the work has been made. If one painting sells for a lot, similar paintings of the same period should also sell well by definition.*

I cannot help but wonder if the effects seen here are basically just "if something is successful, do it again". Peter Jackson makes a second Hobbit movie because the first is successful. If one of my paintings sell, make another just like it and see if it sells. If I am happy with the results of a paper, keep working on it. That would suggest an autoregressive model would generally predict success from the previous works.

Response: First of all, thank you for this insightful and constructive comment.

We fully agree with the Reviewer—the idea “if something is successful, do it again” is indeed a very plausible hypothesis for explaining hot streaks. And, if this hypothesis were true, an autoregressive model that predicts success from the previous works might be a good candidate to describe real careers. Therefore, in this revision, we have spent considerable effort in exploring this possibility suggested by the Reviewer. As we discuss next, testing this hypothesis and the autoregressive model has helped us substantially improve our understanding of the hot streak dynamics while further highlighting the novelty of our results.

We next address this comment in detail. First, we conduct empirical analysis to test the “if something is successful, do it again” hypothesis in the three domains. In doing so, we uncover new evidence regarding the patterns of repetition in successful works, which appear at odds with the proposed hypothesis. Second, we present a quantitative test of

the autoregressive model. We show that an autoregressive (AR) model, while plausible, fails to reproduce patterns observed in data regarding the clusters of hits. Overall, these analyses suggest that while the works produced during the hot streak tend to be similar, they still exhibit a degree of variability not captured by the “if something is successful, do it again” hypothesis. By testing these interesting hypotheses and models raised by the Reviewers, these new analyses have substantially improved our understanding of the underlying processes our paper aims to study.

Testing the hypothesis “if something is successful, do it again”

We fully agree with the Reviewer that “if something is successful, do it again” is a highly plausible hypothesis. The Matthew effect suggests that an individual’s initial success may bring resources and reputation which may help the individual to succeed again in the future. From this perspective, creating subsequent work that is similar to one’s successful prior work may be advantageous, suggesting that one might continue working on a successful style or topic until the competitive advantage dissipates.

We test this hypothesis through one key prediction of the Matthew effect: first mover advantage [1, 2]. It suggests that under the “if something is successful, do it again” hypothesis, we should expect that on average individual performance would decrease over time as one repeats the same recipe for success. Anecdotally, this prediction seems consistent with the many examples, including those quoted by the Reviewer. For example, in terms of IMDB ratings, the first LOTR movie has a higher rating than the second one which has a higher rating than the third one. More systematically, research shows that while sequel films have good box-office performance in general, it is rare for sequels to outperform the predecessors in terms of either gross box office or the probability to earn award recognitions [3, 4]. Similarly, in science, the first study that opens up a new line of inquiry tends to be highly cited. And follow-up studies that continue the investigations may also attract much attention, but are unlikely to be cited at the same level as the canonical paper. Admittedly, individual cases may differ, and there are always exceptions where subsequent works overtake their forebear in prominence. But on average, the Matthew effect would predict that works produced in the early part of the hot streak should have a higher impact than those produced later in the hot streak.

Next, we perform two different measurements to test this prediction: (1) We compare the impact distribution of works produced in the first and second half of the hot streak, as measured by the logarithmic of auction price, the IMDB rating and the logarithmic paper citations in 10 years C_{10} (Fig. R1a-c). (2) We measure the relative position of the highest-impact work among the top six highest-impact works in a career (Fig. R1d-f). Interestingly—and contrary to what the hypothesis would predict—we find no systematic difference between the impact in the first and second half of the hot streak. Moreover, we observe an equal probability for the highest-impact work to appear before and after other hits in a career. In other words, across both measurements, our results show that the impact during a hot streak stays remarkably stable, and does not exhibit decreasing trends in impact. Overall, these results appear in tension with the “if something is successful, do it again” hypothesis. At the same time, these findings improve our understanding of the

hot streak dynamics, further adding to the attractiveness of the underlying phenomena. Indeed, these results suggest that, although individuals become substantially more focused during hot streak, they tend to work on *similar but not identical* topics or styles, further suggesting that people who simply reproduce their past success may not sustain their high-level performance.

Figure R1. (a-c) The impact distribution for the first and the second half of a hot streak across three domains. (d-f) The distribution of the relative position $P(\tilde{N})$ of the three highest-impact works among the six highest-impact works within a career for artists, where \tilde{N} denotes the relative order among the top six hits.

Testing the autoregressive model

Next, based on the Reviewer’s comment, we systematically explore the possibility of using an autoregressive model to reproduce the observed dynamics. An autoregressive model assumes short-range correlations in the impact sequence to predict one’s future impact. Specifically, we can express an autoregressive model $AR(\rho)$ for the impact sequence X_t of each individual career as $X_t = c + \sum_{i=1}^{\rho} \beta_i X_{t-i} + \epsilon$, where ρ is the number of preceding time steps, β_i is the correlation at lag i , c is a constant, and ϵ is white noise.

At the first glance, the autoregressive model is a rather plausible candidate for describing the dynamics of hot streaks. Indeed, as high-impact works are temporally clustered in real careers, correlations should exist in the impact sequence, consistent with the underlying assumptions of the autoregressive model. To test the validity of the autoregressive model, we first fit $AR(1)$ to our data and test whether the model can reproduce the clustering of hits in real careers. We measure the correlations between the relative timing of an individual’s two biggest hits N^* and N^{**} predicted by the best fitted parameters of each individual (Fig. R2a-f). In contrast to patterns in real careers (Fig. R2a-c), we find that there is little correlation between the timing of N^* and N^{**} under the $AR(1)$ model

(Fig. R2d-f). We also calculate the distribution of streak length $P(L)$, as predicted by $AR(1)$, defined as the number of consecutive works whose impacts exceed the median of all works within a career, compared with a null model where the impact sequence is randomly shuffled (Fig.R2j-o). We find that the model again fails to capture the fact that high-impact works tend to be clustered in sequence in real careers. We further relax autoregressive model to allow for larger lags and test the predictions by $AR(5)$, finding again the model fails to reproduce the clustering of high-impact works in real careers (Fig. R2g-i, p-r).

Overall, analyzing the AR model further improved our understanding of the hot streak dynamics. Indeed, these results indicate that while the hot streak dynamics imply a temporal correlation, simply having the temporal correlation by itself is insufficient to reproduce the dynamics observed in real careers. The main reason is that the correlations *vary over time*, according to the hot streak dynamics, which accounts for the period of sustained high performance. Indeed, overall, the level of correlation across a whole career is rather mild. When we measure directly the autocorrelation function for real careers across the three domains, we find that the lag 1 autocorrelation is 0.04 for artists and movie directors, and 0.05 for scientists. The correlation becomes even smaller for larger lags. These results indicate that the assumptions underlying the hot streak model remain the key to reproducing the patterns we observe in real careers; without these assumptions, a generic autoregressive model by itself cannot account for the observed patterns.

Taken together, the alternative hypothesis and the autoregressive model proposed by the Reviewer are highly relevant and insightful, highlighting the Reviewer's deep understanding of the underlying processes we aim to study. Although they cannot explain the intriguing patterns observed in real careers, as we discussed here, these additional analyses not only support the novelty of our results, but have also furthered our own understanding of the underlying phenomenon.

Lastly, the Reviewer's comment on "cause and effect" also prompted us to reexamine the overall language of the paper to avoid causative interpretations. Indeed, the paper presents empirical regularities associated with the onset of hot streaks, and given the observational nature of the study, it does not address causal questions regarding the beginning of hot streaks. We therefore further went through the revised paper to avoid potential causal language.

We thank the review for these rather pertinent comments. In response, we have made several changes in the revised version of our paper:

1. We added a new section in the supplementary information SI S6.7 to discuss the hypothesis of "if something is successful, do it again".
2. We added a new section in the supplementary information SI S6.8 to discuss the autoregressive model.
3. We mentioned in the main text testing of alternative hypotheses with pointers to SI that presents detailed analyses.

Figure R2. Comparison between autoregressive model and real careers. (a-i) The relative timing between the two biggest hits N^* and N^{**} for (a-c) data, (d-f) AR(1), and (h-i) AR(5). We measured the correlation between the N^* and N^{**} , and compared it with a null hypothesis in which N^* and N^{**} each occurred at random. The matrix denotes the value for the normalized joint probability, $\varphi(N^*, N^{**}) = P(N^*, N^{**}) / (P(N^*)P(N^{**}))$. (j-r) The distribution of the length of streaks $P(L)$ of real and shuffled impact sequences for (j-l) data, (m-o) AR(1), and (p-r) AR(5).

3.3 *One important thing left undone is a study of what happens after hot streaks. Do they lead to a resumption of exploration, or is that people stay in the domain trying to hit lightning again? I would encourage the author to report the analysis in the next revision.*

Response: Thank you so much for this insight. Indeed, we were so focused on the beginning of the hot streak, we forgot about the equally interesting question of what happens when it ends. We are grateful for the opportunity to remedy this. Following your insightful suggestion, we first calculate the average entropy $\langle H \rangle$ measured in real careers after the end of a hot streak, and compare them with the randomized careers, measured by the distribution of entropy, $P(\langle H \rangle)$, for 1000 realizations of the randomized careers across three domains (Fig. R3a-c). We find that, after a hot streak ends, $\langle H \rangle$ is statistically indistinguishable from the random expectation, indicating the topic diversity goes back to one's typical level after a hot streak ends. In other words, individuals do not exhibit obvious patterns of exploration or exploitation after the end of a hot streak. Interestingly though, individuals do tend to explore again before the onset of their second hot streak, as we measured for scientific careers in Fig. S23.

We also systematically examine the temporal changes in entropy at the end of a hot streak. Specifically, we align careers based on when their hot streaks end and measure the dynamics of H (Fig. R3d-f) and then repeat the measurement for randomized careers. Before a hot streak ends, H measured in real careers is systematically smaller than expected, consistent with an exploitation strategy. But, the difference between H in the data and in the null model disappears after the end of the hot streak, indicating again that individuals return to their typical level of diversity.

Figure R3. Understanding the end of hot streak. (a-c) The distribution of entropy $P(\langle H \rangle)$ after a hot streak ends for 1000 realizations of the randomized careers for all individuals analyzed in our datasets. The vertical line indicates $\langle H \rangle$ measured in real careers, showing that there is no significant difference between data and the null model. (d-f) The dynamics of topic entropy H surrounding the end of a hot streak for real and randomized careers, measured through a sliding window of six artworks, five films or five scientific papers. Error bars represent the standard error of the mean.

We are very grateful for this insightful comment. We feel that the new analyses presented here offer new insights on hot streaks in careers. Following the Reviewer's excellent suggestions, we have now added these figures directly into the main text, and added new discussions concerning the end of hot streaks in the revised manuscript.

Overall, we are very grateful for the many thoughtful comments the Reviewer shared with us. The additional analyses enabled by these comments have substantially improved the paper. We deeply appreciate your help!

Reviewer #4

4.1 *The paper by Liu et al. studies hot streaks in various types of careers and attempt to understand the patterns that occur prior to the onset of a hot streak. The authors compare the exploration and exploitation of an individual before and during a hot streak, finding that before the hot streak individuals tend to carry out more exploration and during a hot streak they have increased exploitation.*

The results are robust across numerous statistical tests and controls, that the authors carry out. Overall, I believe that the results are very interesting, and the manuscript can be published after the authors consider the following comments.

Response: We thank the Reviewer for highlighting the robustness and interestingness of our findings. Next, we offer a detailed point-by-point response to the rather pertinent comments the Reviewer shared with us.

4.2 *1. While the results are consistent and significant. The effect size of exploration prior to exploitation is not particularly large in many cases. For example, the increase in the likelihood of a hot streak is 10-20% given that one follows the pattern of exploration-exploitation. Similarly, many of the comparisons of the distribution of entropy $\langle H \rangle$ compared to the null model show significant but not necessarily dramatic differences (see e.g. Fig. 3d-3i and Figs. S25-S32). This would suggest that individuals somewhat pare down their focuses during the hot streak, but not dramatically so. Is it possible that within the period of the hot streak there is some control that could be done to highlight the exploitation of one particular area where the individual is having the greatest success yet still continuing, perhaps in a lesser role, in other areas? For scientific careers, this could be assessed by authorship order where perhaps the increased impact occurs in a single area where the researcher is acting as a senior author while still continuing collaborations in other areas? Alternatively, is the increased performance during the hot streak occurring in multiple focus areas simultaneously?*

Response: Here the Reviewer raised a rather thoughtful comment regarding the effect size and suggested several insightful controls to help us further test the effect.

We follow the Reviewer's suggestion and identify, for each scientist, the authorship order in their papers. We approximate the first and last author as lead authorship. We then focus on lead-author publications during a hot streak, and measure again their entropy distribution $P(H)$. As the Reviewer predicted, excluding papers of a lesser role indeed significantly increases the effect size (Fig. R4a, KS-test p -value = 6.1×10^{-9}). Focusing on lead-author publications alone also yields a larger difference in the entropy distribution $P(H)$ before and during a hot streak (Fig. R4b, KS-test p -value = 1.0×10^{-55}).

Figure R4. (a) Cumulative entropy distribution for all papers during hot streaks and the lead-author papers. (b) Cumulative entropy distribution before and during a hot streak for lead-author papers.

In addition to presenting these new analyses which have further increased the effect size, we recognize that readers may have similar questions regarding the effect size. Hence in this revision, motivated by the Reviewer’s thoughtful comment, we added further discussion on this point to help orient the readers. We feel these new discussions further help to clarify the contributions of our paper while helping to direct subsequent research attention for future work. Specifically, we added new discussions on the following points.

First, we recognize that real careers are complex, with heterogeneous influences operating across domains as well as a multitude of individual and institutional factors. While our analysis mainly examines one factor, uncovering significant relationships across a wide range of domains, it may not be the only factor, suggesting fruitful directions for future work. Second, it is interesting to note that the effect size reported here seems on par with other well-known studies that examine the extent to which exploration and/or exploitation are associated with creativity and performance (see e.g. [5] and [6]). Most importantly, the core contribution of our findings is that they unveil among the first empirical regularities underlying the onset of a hot streak. Indeed, prior to this work, the prevailing evidence suggests a random view of hot streaks and individual creativity, characterized by a lack of systematic explanations for the onset of hot streaks. This view is further supported by various analyses on the alternative hypotheses in SI S6, showing the hot streak phenomena resists a number of plausible explanations. From this respect, we feel the findings presented in this paper are both interesting and important, regardless of its effect size, as they reject an important prior, showing instead a close connection between exploration, exploitation and career hot streaks.

Motivated by the Reviewer’s insightful comments, we have made several changes to the paper.

- (1) We have now added the discussions on potential dilution in effect size in the main text.
- (2) We also added a new section in the SI S4.5 and discuss in detail the authorship order and Fig. R4.

- (3) We have added further discussions on the effect size to the revised manuscript, and feel that as a result, these changes have further clarified the contribution of the paper and also help to direct future work in this area.

We thank the Reviewer for these thoughtful suggestions!

4.3 2. *While entropy is a natural metric for measuring the diversity of topics and the level of focus, it is not always easy to interpret. Would it be possible to also add some simpler measures to assess the variety of topics before and during the hot streak? For example, could the authors show what fraction of an author's works are in the most popular topic and what is the distribution of how many topics an author works on before and during the hot streak? These could also be compared to the null model and could help readers' comprehension beyond the entropy metric.*

Response: We thank the Reviewer for another great suggestion. We fully agree with the Reviewer. Both the number of topics m and the fraction of an author's works that are on the most popular topic could be interesting alternative measures to quantify diversity and focus. Next, we present new analyses to test these measures.

Given that the productivity n differs over time, we calculate the average number of topics per paper m/n for papers published before and during hot streaks, and compare to that of the null model. We follow the Reviewer's comment and compare the average $\langle m/n \rangle$ before and during hot streaks measured in real careers to the distribution of $P(\langle m/n \rangle)$ for 1000 realizations of the randomized careers across three domains (Fig. R5a-f). We further directly compare the distribution $P(m/n)$ before and after the hot streak begins for real and randomized careers (Fig. R5g-l). Together, Fig. R5 shows that individuals tend to work on more topics before and fewer topics after a hot streak begins, and this result holds for all three types of careers. Overall, these results are consistent with the shift from exploration to exploitation strategy.

Similarly, we repeat the measurement for the fraction of papers on the most popular topics (Fig. R6), defined as the topic that represents the most works one produces. We find that individuals produce fewer works on the most studied topic before a hot streak, which is consistent with an exploration strategy, but they become more focused on the most popular topic during their hot streak, which is consistent with an exploitation strategy.

We added two new sections in SI (SI S3.11, S3.12) to discuss Fig. R5 and R6 in more detail. We also modified the main text to add pointers to these analyses.

Figure R5. Repeat the measurement for the number of topics per paper. (a-f) The distribution of the average number of topics $P(\langle n/m \rangle)$ before and during a hot streak for 1000 realizations of the randomized careers for all individuals analyzed in our datasets. (g-l) Cumulative distribution $P(\leq m/n)$ before and during a hot streak in real and randomized careers across the three domains.

Figure R6. Repeat the measurement for the fraction of papers in the most popular topic. (a-f) The distribution of the average fraction $P(\langle fraction \rangle)$ before and during a hot streak for 1000 realizations of the randomized careers for all individuals analyzed in our datasets. (g-l) Cumulative distribution $P(\leq fraction)$ before and during a hot streak in real and randomized careers across the three domains.

4.4 3. Regarding the sizes of teams, the authors only explored this area for scientists. I realize that artists work alone and thus there is assessing team size there, however, is it possible to assess team size for film directors since films typically result from many individuals? Even if films are more fixed in their overall team size due to the many parts involved, it would at least be interesting to show a graph similar to Fig. S36 for films. In this context also seem relevant the recent study of An Zeng et al arXiv:2007.05985 which shows the effect of new collaborators.

Response: We thank the Reviewer for this important comment. We fully agree with the Reviewer that a film is also a team product. To clarify, in the original manuscript, we analyzed the team size for science not because it is the only domain with data. Rather, it is motivated by the recent literature [7], which shows that in science, large teams excel at solving problems, and small teams tend to be more disruptive and are more likely to come up with new questions and opportunities. The documented relationship between team size and research outcomes in science prompted us to examine the change of team size in scientific careers. To the extent team size may be a relevant construct for film directing, its theoretical motivation remains unclear. Indeed, it remains unclear if and how the team size may condition the character of the work, hence unclear if it is relevant to the exploration and exploitation strategy discussed in our paper. Meanwhile, there are also reasons to believe team size may not be the right construct in film directing. A film typically involves more than 60 team members, much more than the typical team size of a paper. Both crews and casts may influence the way a director produces films. And, the team organization and workflow may differ substantially from the nature of collaboration among scientists. Following the Reviewer's comment, we revised the team paragraph in the manuscript to clarify the motivation of the team size result, and why we only focused on science in this case.

We also thank the Reviewer for pointing us to the highly relevant and interesting study by Zeng et al [8], which shows that teams involving more new collaborators tend to produce papers of higher originality and more multi-disciplinary impact, especially for large teams. These findings are consistent with several of our findings (see for example Fig. 4 and SI S4.3), offering a new perspective to understand the role of scientific teams in knowledge production. We now cite and discuss this paper in the main text and SI S4.3.

4.5 4. Regarding the team sizes, can the authors add error bars to the graphs in Fig. 4, or how large is the deviation in team sizes on the various points?

Response: Absolutely. We have now revised the paper to show error bars. We also shrink the marker size in Fig. 4a and make the error bars more readable (see Fig. R7a). We also calculate the variance for Fig. 4b with bootstrapping and add error bars to Fig. 4b in the revised manuscript (see Fig. R7b).

We have replaced Fig.4 with Fig. R7.

Figure R7. Team size in science (a) The average team size around the beginning of a hot streak for real and randomized careers in science. (b) We calculate team size for papers published before and during hot streak, and compare the distribution to that of randomized careers for 500 realizations, denoted as $R(\text{team size})$. Both measures in a and b suggest that scientists tend to engage with smaller teams before hot streak, and with larger teams during hot streak.

4.6 5. *While the hot streak model is based on prior work, can the authors justify in the SI the use of two normal distributions for representing the typical performance and elevated performance? Generally, performance in these types of domains tends to have a long tail (power-law), and thus would it not make more sense to consider for example two power-law distributions?*

Response: We thank the Reviewer for highlighting the fat-tailed nature of the underlying distributions. The Reviewer is correct that both the raw auction price and paper impact follows fat-tailed distributions, which can be approximated by a log-normal distribution. This is why in our preprocessing of the data, we have taken the logarithm of auction price and paper impact. Hence the normal distributions used to represent performance measures are precisely meant to capture the fat-tailed nature of the raw measures. We wish to thank the Reviewer for helping us realize that this important point has not been discussed properly in the paper. We added discussions in the main text and SI S1.5 to highlight the fat-tailed distribution of the auction price and citations and the fact that we have taken the logarithmic of these impact measures.

Other minor points:

4.7 1. *On page 5 in the paragraph beginning “To systematically examine...” the phrase “during hot streak” should be “during a hot streak”.*

2. *In the caption to Fig. 1 in the last sentence, “we apply community detection algorithm” should be “we apply a community detection algorithm”.*

Response: We thank the Reviewer for these comments. We have changed these

sentences accordingly in the revised manuscript.

4.8 3. *In Fig. 3 is it possible to either separate the x-axis labels more? Currently, it is difficult to understand how to read the labels that are on two lines.*

Response: We thank the Reviewer for pointing out this issue. We have adjusted the x-axis labels in Fig. 3s-u in the revised manuscript to make them more readable.

Thank you again for all your constructive comments and suggestions. They have inspired many new analyses that we believe have substantially strengthened the paper. Please do not hesitate to let us know if there is anything else we can do to further improve this work!

Reviewer #5

5.1 The main contribution of this work is to apply advanced computational methods of deep learning and network science to build high-dimensional representations of creative works grouped into career trajectories, thereby affording analysis of the transition into a hot streak. On this merit alone, the manuscript deserves consideration.

Response: We wish to thank the Reviewer for highlighting the novelty of the paper in its use of advanced computational approaches. Indeed, an important contribution of our paper is to help to illustrate that recent advances in deep learning and high-dimensional embedding may enable us to analyze creative works at a new level of scale and detail, opening up exciting new ways to quantitatively probe patterns related to creativity. In this work, these advanced computational methods allowed us to address one of the main open questions in the field, namely, empirical regularities associated with the onset of hot streaks. More generally, we also hope that our methodology as well as our findings can pave the way for many exciting new works to come that would together deepen our understanding of the many complex social systems. Next, we offer point-by-point responses to each of the insightful comments the Reviewer shared with us.

5.2 However, upon a close read, many conclusions are presently based upon the argument that consistent results across the three domains somehow is justification in of itself. It would be more satisfying to see results that differentiate between the domains that are consistent with fundamental differences in the role of exploration, exploitation and success in each.

Response: Thank you for this helpful comment; it helped us to realize that an important aspect of our results has thus far been overlooked by us. We are grateful for this insightful comment and glad that this revision offered us a chance to remedy this. Indeed, on the one hand, we are amazed by the fact that our findings appear rather universal across the rather diverse domains we studied. Given the diversity and complexity of these domains, it is conceivable that the patterns underlying the beginning of hot streaks may differ across domains. From this respect, we were quite intrigued that not only is the phenomenon of hot streaks quite universal across domains, as documented in the prior work, the regularities associated with the onset of hot streaks also appear quite reproducible. On the other hand, the Reviewer's comment also made us realize that while we emphasize the universality of the results in the original manuscript, we have also overlooked the important domain-specific patterns. Indeed, individual success and creative strategies are influenced by complex, heterogenous, and domain-specific factors related to institutions, cultures and more. We fully agree with this and believe that understanding how patterns may differ across domains is an important question.

Following the Reviewer's suggestions, we look deeper into the differences across domains and further highlight these results in the revised manuscript. First, we

investigated the number of styles/topics within each career. We find that the overall distribution varies across domains (Fig. R8), which highlights important domain-specific differences. Further, we examine how patterns of exploration and exploitation may differ across domains. We find that the level of exploration and exploitation in science appears much stronger than in art and film industry (Fig. 3). For example, the z-score of $\langle H \rangle$ for science is 13.9 before hot streaks and -22.7 during hot streaks, about three times the value compared with artists and directors. Overall, these preliminary results show that, beneath the universal associations we document between exploration, exploitation, and the onset of hot streaks, there are indeed interesting differences across domains, as the Reviewer suggested. Attending to these domain-specific patterns has further strengthened the paper, which now presents a richer and more balanced narrative with both universal and domain-specific patterns. We are grateful for this insightful comment by the Reviewer.

At the same time, given the heterogeneity in the domains we study and the methods to study them, we also want to hew closely to our conservative instincts in interpreting these cross-domain differences, as these patterns could also be attributed to inherent differences in data and methods, in addition to differences in the role of exploration, exploitation and success. Hence, we also recognize that decoding the roots of these domain-specific patterns remains a challenge, and in this revision, we also added explicit calls for future work into this important direction.

Following your suggestion, we added a new paragraph in the main text to highlight these domain-specific patterns and also explicitly call for more future work on systematically assessing the fundamental differences in the role of exploration, exploitation and success.

Figure R8. The distribution for the number of topics $P(m)$ within a career for the three domains.

Detailed comments:

5.3 *The authors focus on March’s exploitation-exploration dichotomy to resolve some of the open questions identified in their parent paper “Hot streaks in artistic, cultural, and scientific careers” (Liu et al, Nature 2018), where they reported that “The hot streak emerges randomly within an individual’s sequence of works, is temporally localized, and is not associated with any detectable change in productivity.” Here they elaborate on this prior work by analyzing shifts in creative exploitation-exploration strategy occurring at the onset of hot streaks.*

Response: Thank you for an accurate summary of the manuscript.

5.4 *The authors rely on statistical comparison with randomized data to arrive at most of their conclusions. Yet there are more sophisticated means of identifying causal effects, and so it’s not clear why they resort to distribution-level comparisons with shuffled data and the like. As elaborated below, it’s also not clear how the authors account for biased estimates arising from highly correlated observations.*

Response: Thank you for this important comment. The choice of using a “null model” to represent our data is rooted in the recognition that the dynamics of individual careers are complex, influenced by a multitude of observable and unobservable factors. From this respect, randomization is a natural choice. The distribution-level comparison between real careers and the null model allows us to extract statistically meaningful patterns while controlling for other potential factors. We also wish to clarify that in this paper we do not attempt to make any causal claims. The primary goal of the paper is to investigate empirical regularities associated with the onset of hot streaks. Future work using causal research designs may offer causal interpretations on the regularities reported here. Following this important comment, we went through the revised paper to further make sure that our writing does not lead to causative interpretation of our findings. We also added a discussion in the revised manuscript clarifying that our results do not have causal implications.

Here the Reviewer alluded to his/her additional questions regarding the biased estimates related to correlated observations, which we will address in 5.10 when this issue is further elaborated by the Reviewer.

5.5 *One of the main results highlighted in the abstract is “a particular sequence of exploration followed by exploitation, where the transition from exploration to exploitation closely traces the onset of a hot streak”. While at first this statement may cause pause for reflection, it’s not clear in what scenario one would expect the reverse, exploitation followed by exploration, to be a reliable alternative strategy. If exploration sets the conditions for arriving at novel creations, then it seems that a better summary of the results is that the onset of hot streaks are marked by more novel creations, which again is not surprising. It appears as though the hot streak is more of a doubling-down on intermittent success more than randomly arriving at King Midas’ golden touch.*

Response: Here the Reviewer commented that our main result ‘transition from exploration to exploitation closely traces the onset of a hot streak’ seems rather plausible, especially when compared to the reverse—exploitation followed by exploration. We fully agree with the Reviewer. Indeed, we are very much encouraged by this comment, as it further supports the validity of our findings. Yet at the same time, it also made us realize that we failed to properly contextualize these findings in the literature which our study contributes to and builds upon. We are delighted to elaborate below.

Given the literature, there are four different strategies that are potentially relevant: (1) pure exploitation around the onset of the hot streak, (2) pure exploration, (3) exploration followed by exploitation, and (4) exploitation followed by exploration. We conducted a systematic review on trade-offs between exploration and exploitation and their relationship to creativity and learning, surveying around sixty studies spanning over fifty years (see SI S2.2 and Table S1). In surveying the literature across several different fields, we find that although extant literature has documented the fundamental role of exploration and exploitation in creativity and performance, the vast majority of literature focus on the first two strategies and their benefits and downsides, while paying (surprisingly) little attention to the sequential combinations of exploration and exploitation. Curiously, in this paper, we find that, across a wide range of creative domains, a major turning point for individual careers appears most closely linked with neither exploration nor exploitation behavior in isolation, but rather with the particular sequence of exploration followed by exploitation, which highlights one of our main contributions: our results highlight that a sequential view of creative strategies that balance experimentation and implementation may be particularly powerful for producing long-lasting contributions. We believe this finding is novel at two different levels:

First, at the theoretical level, while this sequential view of creative strategies may seem intuitive—once we discovered it from data—it has received little attention in the literature, especially related to career-level analyses (see S2.2). We therefore hope that this finding will contribute to the extant literature on exploration and exploitation, and further this rather foundational school of thought by suggesting that one might consider the sequential view of exploration and exploitation rather than consider them in isolation or in combination.

Second, at the empirical level, this finding is also novel in the context of existing large-scale analyses of creative careers. For example, while the reverse scenario of exploitation-exploration might appear unproductive, as the Reviewer suggested, recent key findings on scientific careers [9] suggest that it indeed seems to be the kind of dynamics that researchers tend to follow: scientists are more focused in early career stages, and feature increasing exploration over time. Indeed, as we discuss in the paper, in science, there are various forces that currently appear in tension with the exploration-exploitation dynamics uncovered in our paper, ranging from the pressure to publish to tenure evaluation favoring focused contributions. There is also widespread evidence that exploitation as a conservative strategy may limit individuals' ability to consistently produce high-impact work and incur negative consequences in the long run [10-13], which again runs counter to the hypothesis that the hot streaks are associated with an exploitation strategy.

Taken together, this thoughtful comment by the Reviewer highlights precisely the gap in the literature that our paper attempts to fill. We are very grateful for this comment as it made us realize that our core contribution could have been better highlighted. We have now edited the discussion section in the revised manuscript, specifically highlighting the contribution of our results and aiming for greater clarity. As a result, we feel the paper has improved substantially in its clarity and its appeal to broader audiences.

The Reviewer commented on the differences between capitalizing on intermittent success vs King Midas' touch. This point was further elaborated by the Reviewer in 5.10. Hence we will discuss this point in detail in 5.10 with several new analyses.

5.6 This consideration brings forth a bigger question of survival bias regarding the many explorers who were ejected because they did not identify the 'right' novel creation before their time to capitalize on the creation evaporated. This point should also be considered regarding the selection of visual arts analyzed from museum and art gallery collections, which depend on strict economically-motivated selection criteria. What about the art works that were not shown? Would one expect them exhibit the same exploration-exploitation patterns as the exhibited work? This would provide more convincing evidence that the source of the strategy shift has its origins with the creator. Recent research by Balietti et al. (Peer review and competition in the Art Exhibition Game, PNAS 2016) shows how competitive exhibition can alter the perceived evaluation of creative works. So do creators transition into the more lucrative exploration state as a result of their own inclination or in response to the external market? Or in terms of the author's statement "On the other hand, the sequence of exploration followed by exploitation may facilitate the emergence of high-impact work by incorporating new insights into a focused agenda", is the agenda driven by the artist or the artgoers? Or in the context of the collaboration sizes, is the hot streak driven by the scientist or her new collaborators? In this regard, the analysis has the effect of raising more questions than it answers.

Response: We thank the Reviewer for these stimulating comments. Here the

Reviewer raised two main points: (1) potential survivorship bias in data, especially regarding many explorers might have been ejected; and (2) the observed exploration-exploitation patterns may be driven by different forces—e.g. it could be due to the market or the individual.

First, we agree with the Reviewer that the individuals in our datasets are “survivors” with long enough careers in each domain. This research design is partly by choice and partly by necessity. Indeed, since our study aims to probe empirical regularities associated with a phenomenon reported in the previous paper [14], it is important to be consistent with the research design of that previous study so the patterns we observe here are directly comparable across the literature. As for the previous study, it seems necessary to have focused on careers that are long enough, as otherwise, it is difficult to define and measure ‘hot streaks.’ Nevertheless, we fully agree with the Reviewer that given the data we have, we cannot eliminate the potential survivorship entirely, as individuals in the left tail of the distribution may have been filtered out before the analysis, and famous works may be overrepresented in museum exhibitions. In this study, we are cognizant about this point, and we have also tried to mitigate it somewhat. For example, we also collected data from online art database Artnet, which contains art images from dealers such as galleries and auction platforms. Our hope was that, by combining images from the two datasets, it would help us construct a more comprehensive (and less biased) profile for artistic careers, especially to capture the less famous works an individual produced. Hence the reported exploration-exploitation patterns in the paper do include both the exhibited works during an artist’s famous period as well as non-exhibited works from the rest of his/her career. But still, potential biases may persist. And given the data-driven nature of our study, this is not something that we could address conclusively. Therefore, recognizing this important point, in this revision, we have added a separate paragraph to discuss this issue, acknowledge this issue clearly in the main text as an open question, and hope to bring this issue to the attention of the research community.

That being said, we also wish to point out respectfully that, through these discussions, it is easy to forget that the three datasets we used in this paper, while far from perfect, are nevertheless to our knowledge among the best and largest datasets currently in existence to study large-scale career histories and outputs, offering the patterns of productivity and impact in an unprecedented level of scale and details. In other words, to the extent the potential survivorship bias is an issue in the data, it is by no means specific to our study; rather it applies to *all* studies in this kind of analyses. And in this paper, we were fortunate enough to overcome the many computational barriers to compile and analyze creative careers in three different but important domains, striving for a high degree of consistency across domains. Hence, from this perspective, the datasets we used here, while not immune to potential bias, are in fact the strength of our paper, not weakness, and part of the reasons why we feel our paper is ideally suited for *Nature Communications*.

Following this comment, we added discussions on data limitation in the revised

manuscript, and added a new section in SI S1.4 to articulate its limitations as well as highlight its novelty and contributions.

The Reviewer's second question asks whether the strategy shift is driven by the individual or other external factors. This is another excellent question. We thank the Reviewer for pointing us to the highly relevant paper by Balietti et al [15], which we now cite in the revised manuscript. We fully agree with the Reviewer that transition from exploration to exploitation raises interesting questions regarding the reason behind the shift. As [15] shows, individual's creative strategy is likely to be influenced by many external forces. Prior literature also suggests that new collaborations [8], social network structure [16], institutional context and disciplinary culture [10, 11] are all plausible factors. Furthermore, individuals may receive short-term feedback (e.g. art critiques or peer reviews) that offers additional signals shaping their career focus. Given the complexity and heterogeneity of career dynamics we examined, our analysis serves as an initial step to understand the problem. From this regard, we are in fact delighted to hear that the Reviewer considered this analysis may raise a series of new questions hence prompt more future follows in this research program. Indeed, the goal and contribution of our paper is to report these new, interesting, and highly generalizable facts. And it is important to recognize that the observed association between hot streaks and exploration-exploitation dynamics holds true, regardless of whether it is driven by individual or external factors. At the same time, we also hope that by documenting these highly generalizable facts, this work can inspire future research dissecting the underlying forces behind these facts. Indeed, we ourselves are also planning to test some of these potential factors going forward, as we hope others will too. In other words, we can debate about the reasons, which we and many others will continue to investigate, but it does not diminish the importance of these new facts.

Following the Reviewer's insightful comments, we added specific discussions on individual vs external factors in the revised manuscript, explicitly calling for future work. We want to thank the Reviewer for this important comment which helps to shape the research efforts from the community.

5.7 In summary, this work provides an almost overwhelming body of evidence that is counter to what was originally reported, that hot streaks are not associated with any detectable change in productivity; the authors are now repositioning hot streaks as being clearly associated with changes in productive strategy as they relate to team size. These new results would indicate that the hot streak has more to do with the market conditions for success than the individual creator.

Response: We thank the Reviewer for this important comment, which makes us realize that the argument 'hot streaks are not associated with any detectable change in productivity' from [14] may seem ambiguous to readers when presented by itself. A more precise statement in that context is that the number of works produced during hot streaks is not significantly different than expected by the null model in which a hot streak appears at random. This argument comes from the comparison between the

distribution of number of artworks/films/papers $P(N_H)$ produced during hot streaks in real careers, and that predicted by a null model in which we randomly pick one work in a career and designating its production year to be the start of the hot streak. We found no detectable differences between $P(N_H)$ in data and the null model. Hence the previous work refers to publication *volume* not *strategy*. In other words, our findings here are not in conflict with what was originally reported; rather, they move those results one step further by identifying new regularities that are consistently associated with the onset of hot streaks.

In a similar spirit, the new results on team size also represent new understandings that take us one step further from the original 2018 paper [17]. Indeed, this team size result was only possible thanks to a subsequent paper published in 2019 [7], which shows that large teams tend to develop science and technology and small teams tend to disrupt it with new ideas and opportunities. This result then makes us wonder if we can use team size to further interrogate our findings. Indeed, if, as our findings suggest, individuals tend to explore new directions before hot streak, and focus on specific directions during hot streak, then one would expect that works that precede hot streaks are more likely to be done with small teams (disruptive), and those during hot streak are more likely to be done with large teams (developmental). And this is indeed what we find, which should be considered as supporting evidence for our findings. In retrospect, this team size result would not have been possible for us to obtain in our original 2018 paper [17], as without the 2019 paper on small teams [7], we would not have known that team size could be a neat and important construct to understand the organization of innovative activities. Therefore, as the Reviewer correctly suggested, these new results do start to show changes in productive strategies associated with hot streaks, even though the sheer volume of paper productions appears largely unchanged during the hot streak. In other words, these new results don't contradict what was reported previously; rather they highlight the kind of new advances that our paper presents in moving the literature forward with new facts and hypotheses.

We wish to thank the Review for this comment, suggesting that our readers may have similar questions when reading the paper. To avoid potential confusion, we have now added more discussions in the main text to highlight these important new advances. As a result, we feel this revision has tightened our paper's relationship with prior literature. We are grateful for the Reviewer's comment.

Additional comments:

5.8 A) *It's not clear why exploitation would consist of in film directing that would be measurable given the data collected. The motivation and definition of what exploitation consists of in each profession is not clear. The authors seem to take it as a given, and so the rhetorical statement "Are career hot streaks reflective of exploration or exploitation behavior, or some combination of the two?" seems self-fulfilling.*

Response: We thank the Reviewer for this comment, which made us realize that we have not explained clearly some of our key measurements. To clarify on these points, we will first discuss the motivation and definition of exploitation in each profession. We will then discuss the measurement of exploitation in film directing specifically. To avoid repetition, we will discuss the seemingly self-fulfilling hypothesis in 5.11 where it was further elaborated by the Reviewer.

Exploitation in art reflects paintings that are similar in style. e.g. the artworks may contain similar objects, convey similar themes, or use similar skills/techniques. This could correspond to famous examples such as van Gogh's series of sunflowers, Picasso's blue period and its theme of poverty and despair, Warhol's repetition in the use of objects, and more. Exploitation in film directing reflects a pattern of making similar films in succession, which may share similar story lines and/or characters. Film sequels are typical examples: they share similar stories with largely overlapping characters and casts, and they typically belong to the same genre. By contrast, exploration corresponds to films with diverse styles as well as different stories, characters, and genres. Similarly, exploitation in science involves papers on similar research topics or using similar methodologies or techniques, whereas exploration engages experimentation on diverse ideas and areas of research. Overall, these measurements are motivated by decades of research on exploration and exploitation in creativity related domains, as reviewed in SI S2.2, which shows that exploitation allows individuals to practice skills and accumulate experiences in a particular area. This in turn fosters recognition and reputation related to the expertise. It thus suggests that exploitation could be relevant to the hot-streak dynamics, for it helps individuals to achieve greater efficiencies while developing a focused agenda.

To quantify the characteristics of film directing, here we focus on the director of each film, as they are commonly considered to play an outsized role in shaping the film's creative vision and success. Indeed, a director's responsibility for a film influences nearly all aspects of a film, from pre-production to approving the final edit. Importantly, directors are responsible for working with and overseeing scriptwriters as they work on the script as well as selecting and training the cast. Given the director's role in shaping these elements of the film, we use data on plot and casting to analyze each film.

Specifically, we collected the plot and cast information for each film and quantify films' similarities by combining the two features. We learn the word embedding from the plot to distinguish different language topics and sentiments across films. We also learn node embedding from a co-casting network to identify films sharing similar actors and actresses. The plot and actor networks capture the genre and theme of the film. We tested this by combining the two representations and using a 200-dimensional vector to predict the genre (e.g., horror, romance) of each film (SI S1.2). Although we did not utilize any genre information to learn the film embeddings, we find that they can predict the film genres with a high accuracy of 0.948. It therefore suggests that, our measures of exploitation and exploration relate to genre, theme and other patterns reflected in the embedding of the plot+cast combination. Exploitation

reflects films close to each other within a cluster, and exploration reflects films located in distant positions, belonging to different clusters on the high-dimensional embedding space.

Following the Reviewer's comments, we have now added a new section in SI S1.6 for detailed discussions on the meaning of exploitation and exploration for all professions in this revision. As a result, we feel the paper has improved greatly in its clarity.

5.9 B) The authors use entropy measures to quantify exploration versus exploitation. This is an odd choice, albeit convenient, as it measures diversity which does not necessarily distinguish exploration from exploitation. An individual may explore a single domain by going deeper. A metric that captures transitions between topics would better capture the phenomena being analyzed. Another issue is that the number of topics m is a variable that may vary between domains, but there is little discussion of how this impacts the results. Fig S14, 17 and 18 showing the distributions $P(H)$ indicate that the location of H values are highly variable. On closer inspection, the difference between the location of the null line and the bulk of the $P(H)$ distributions are relatively small also.

Response: We thank the Reviewer for these thoughtful comments. Here, the Reviewer made three suggestions: 1) use transition between topics to capture exploration and exploitation; 2) discuss the number of topics m across domains and how it impacts results; and 3) discuss the relatively modest effect size. These are all great comments. We next address each one of them in detail.

(1) Transition between topics. We agree with the Reviewer that capturing transitions between topics offers an interesting alternative to capture the phenomena being analyzed. Following the Reviewer's suggestion, we measure the probability that an individual switches style or topic between consecutive works for periods during and before hot streak. We find that across all three domains, individuals are more likely to switch topics before hot streaks and less likely to do so during hot streaks, which are consistent with our overall conclusions (Fig. R9). We agree with the Reviewer that this new analysis well captures the phenomena being analyzed, which provides further support for our results. We now mention this result in the main text, and added a new section in SI S3.13 to discuss Fig. R9 in more detail.

Figure R9. Repeat the measurement for probability of switching topics between two consecutive works. (a-f) The distribution of the average probability of switch $P(\text{probability of switch})$ before and during a hot streak for 1000 realizations of the randomized careers for all individuals analyzed in our datasets. (g-l) Cumulative distribution $P(\leq \text{probability of switch})$ before and during a hot streak in real and randomized careers across the three domains.

(2) Discussions on the number of topics. The Reviewer is also right that the number of topics varies across domains. Indeed, following the Reviewer’s comment, we examine the differences in the number of topics across the three domains, and compare the distribution for number of unique styles/topics within a career (Fig. R10). Consistent with the Reviewer’s insights, the number of topics m indeed varies across domains: the median m is 7 for artists, 3 for directors and 6 for scientists. It is important to note that the differences in topic numbers do not affect the entropy results reported in the paper, since our null model randomizes within each career. Hence the topic entropy result represents a *within-person* comparison, which allows us to account for the topic

number differences across domains and individuals. This indicates that the reported exploration-exploitation transition holds regardless the domain differences in the number of topics. Nevertheless, we fully agree with the Reviewer that the number of topics m and how it impacts the results are important questions and deserve more attention in the paper.

Following the Reviewer's insight, we dig deeper into the number of topics, and present three levels of analyses that we feel have further strengthened the paper.

First, we consider the heterogeneity of m across domains and individuals, and use regression analysis to specifically control for the number of unique styles/topics. We find that our conclusions remain the same (SI S3.5).

Second, to further test the robustness of our results over m , we conduct an alternative empirical test by measuring the topic entropy for only individuals with similar m in each domain. Specifically, we control for individuals with m around the median value (6 to 8 for artists, 2 to 4 for directors, and 4 to 6 for scientists). We then compare the average entropy measured in these careers to 1000 realizations of the randomized careers. (Fig. R10a-f). We also directly compare the distribution of entropy before and after the hot streak begins for real and randomized careers (Fig. R10g-l). All these results show that individuals tend to work on more diverse topics before a hot streak and become more focused after a hot streak begins, suggesting that the shift from exploration to exploitation strategy still holds after we control for m .

To take these ideas even further, we next directly use the number of styles/topics as a proxy for exploration/exploitation and repeat our analyses. We expect to observe more topics before hot streak and fewer topics during hot streak. Given that the productivity n differs over time, we normalize this effect by calculating the average number of topics per unit of production m/n for works before and during hot streaks, and compare to that of the null model. We compare the average $\langle m/n \rangle$ before and during hot streaks measured in real careers to the distribution of $P(\langle m/n \rangle)$ for 1000 realizations of the randomized careers across three domains (Fig. R11a-c). We further directly compare the distribution $P(m/n)$ before and after the hot streak begins for real and randomized careers (Fig. R11g--l). Across all these measurements and all three different domains, we again find consistent results. Indeed, Fig. R11 shows that overall individuals tend to work on more topics before and fewer topics after a hot streak begins, consistent with the shift from exploration to exploitation strategy. _

We added a new section in SI S3.11 to discuss Fig. R10 and Fig. R11 more in detail.

Figure R10. Robustness test for the number of topics. We repeat the measurements only for artists with 6-8 styles, directors with 2 to 4 styles and scientists with 4 to 6 topics.

Figure R11 Use the average number of topics per work to quantify exploration and exploitation, and repeat the analysis across three domains.

(3) Discussions on the modest effect size. The Reviewer raised another important point regarding the effect size. We fully agree with the Reviewer, and this insightful comment made us realize that this point requires more discussions in the paper. Indeed, real careers are undoubtedly complex, with large heterogeneity across individuals and domains, influenced by a multitude of individual and institutional factors. Although here we mainly focused on one potential factor, namely the effect of exploration and exploitation, it doesn't mean it's the only factor at work. Future work can extend the analysis to examine other factors that may also play a role, and assess how multiple factors may interact with each other. Indeed, we hope that our paper will inspire future studies to further examine other various factors that might also contribute to the onset of hot streak.

That being said, the Reviewer’s comment made us wonder if there are further controls that we can do to better isolate the effect and further increase the effect size. Here we carry out a new analysis that separates a scientist’s main topics and collaborative topics. The hypothesis is that while an individual may achieve great success in some topic during hot streak, one may still continue publishing on other topics that started before the hot streak, perhaps in a lesser role. Hence if we control for this factor, we should expect the effect size to magnify. This hypothesis was also raised by Reviewer 4, who suggests that “[f]or scientific careers, this could be assessed by authorship order where perhaps the increased impact occurs in a single area where the researcher is acting as a senior author while still continuing collaborations in other areas” To test this hypothesis, we identify for each scientist the authorship order in their papers. We approximate the first and last author as lead authorship. We then focus on lead-author publications during a hot streak, and measure again their entropy distribution $P(H)$. We find that excluding papers of a lesser role indeed significantly increases the effect size (Fig. R12a, KS-test p -value = 6.1×10^{-9}). Focusing on lead-author publications alone also yields a larger difference in the entropy distribution $P(H)$ before and during a hot streak (Fig. R12b, KS-test p -value = 1.0×10^{-55}).

Figure R12. (a) Cumulative entropy distribution for all papers during hot streaks and the lead-author papers. (b) Cumulative entropy distribution before and during a hot streak for lead-author papers.

In addition to presenting these new analyses which help to highlight the effect size, we recognize that readers may have similar questions regarding the effect size. Hence in this revision, motivated by the Reviewer’s thoughtful comment, we added further discussions on this point to help orient the readers. We feel these new discussions further help to clarify the contributions of our paper while helping to direct subsequent research attention for future work. Specifically, we added new discussions on the following points.

First, we recognize that real careers are complex, with heterogeneous influences operating across domains as well as a multitude of individual and institutional factors. While our analysis mainly examines one factor, uncovering statistically

significant regularities across a wide range of domains, it may not be the only factor, suggesting fruitful directions for future work. Second, it is interesting to note that the effect size reported here seems on par with other well-known studies that examine the extent to which exploration and/or exploitation are associated with creativity and performance (see e.g. [5] and [6]). Most importantly, the core contribution of our findings is that they unveil among the first empirical regularities underlying the onset of hot streaks. Indeed, prior to this work, the prevailing evidence suggests a random view of hot streaks and individual creativity, characterized by a lack of systematic explanations for the onset of hot streaks. This view is further supported by various analyses on the alternative hypotheses in SI S6, showing the hot streak phenomena resists a number of plausible explanations. From this respect, we believe the findings presented in this paper are both interesting and important, regardless of the effect size, as they reject an important prior, showing instead a close connection between exploration, exploitation and career hot streaks.

Following the Reviewer's comment, we have now added 1) the discussions on potential dilution in effect size in the main text, 2) a new section in the SI S4.5 to discuss Fig. R12 in detail, and 3) further discussions on the effect size in the main text. We feel that as a result, the paper has further clarified its contribution and also helps direct future work in this area.

5.10 C) *Are the authors measuring hot streaks in which creators have the King Midas touch, or are they just capitalizing on variations around a successful theme? Peter Jackson's career exhibited in Fig 2 provides a good example, where it appears as though the Lord of the Rings episodes are treated as 3 independent observations? The same could be argued for all paintings belonging to Pollock's drip period - are these to be considered separate or variations on the same theme? How much do these types of correlated observations explain the main results of the analysis?*

Response: We thank the Reviewer for this insightful comment. It is also related to Comment 5.4: '*As elaborated below, it's also not clear how the authors account for biased estimates arising from highly correlated observations.*' and Comment 5.5: '*It appears as though the hot streak is more of a doubling-down on intermittent success more than randomly arriving at King Midas' golden touch.*'. Here we combine our responses to these thoughtful comments.

First, the Reviewer noted that the drip period paintings or Lord of the Rings films are from the same theme. One potential worry here is that this may lead to biased estimation if we purposefully assign works during an individual's hot streak into the same theme, in which case the exploitation patterns observed may be an artifact as opposed to revealing the underlying phenomenon. However, it is important to note that here we only cluster the works based on their content, not based on whether or not they were part of the hot streak. Indeed, as we only feed the image, film plots and casts into the neural networks to learn their representations, the models are agnostic about the authors, or the timing when the work was produced, or whether it belongs

to a normal phase or a hot streak. As such, the models do not expect works created during a hot streak to belong to the same style. Hence the highly correlated observations represent *new patterns from the data, not artifacts* of the measurements. In other words, the fact that similar works coincides with hot streaks should be viewed as empirical support for the exploitation strategy we discovered.

Further, the Reviewer raised another insightful comment: Our results seem to suggest that, “more than randomly arriving at King Midas golden touch”, hot streak appears “more of a doubling-down on intermittent success” (“capitalizing on variations around a successful theme”). Indeed, given the randomness in when hot streak occurs, it would appear as if individuals randomly arrive at a kind of King Midas golden touch. This underscores a key contribution of our paper. Indeed, in our paper we show that exploration followed by exploitation is consistently associated with the onset of hot streaks, and there are close connections between the topics that were exploited during hot streak and those explored before the hot streak arises. Hence a key contribution of these results is to show that the onset of hot streak is not simply magical. Rather, there are consistent empirical regularities associated with it. It then begs an intriguing question that the Reviewer shared with us—is it just capitalizing on an initial success and doubling-down on that success? We are fascinated by this question, and in this revision we have spent considerable effort in exploring it. We will discuss our new analyses in detail below. Overall, these new analyses add important nuances to the overall picture. As such, testing this hypothesis has substantially improved our understanding of the hot streak dynamics while further highlighting the novelty of our results.

We fully agree with the Reviewer that the hypothesis of doubling-down on intermittent success is indeed highly plausible. The Matthew effect suggests that an individual’s initial success may bring resources and reputation which may help the individual to succeed again in the future. From this perspective, creating subsequent work that is similar to one’s successful prior work may be advantageous, suggesting that one might continue working on a successful style or topic until the competitive advantage dissipates.

Although this hypothesis shares high-level similarity to the exploration followed by exploitation pattern we documented, we find that there are two sources of evidence that add important nuances to the overall picture.

First, the doubling-down on intermittent success hypothesis offers important predictions on what kinds of topics tend to be exploited during hot streak, among those that were explored before. For example, one might expect that the exploited topic may be the one explored the most recently, as individuals may switch to exploitation once they stumble upon a promising direction. It is also likely to be the highest cited or the most popular topic among those explored before. We can test these properties from data. But we find that the exploited topic is in fact *less* likely to be the most recently explored, or the highest cited, or the most popular among the topics explored before (see SI S5 and Fig. S44). These results seem to run counter to the “doubling-down on intermittent success”

hypothesis. Rather, they imply that individuals appear to deliberate over different possibilities before they decide which direction to pursue.

Second, we carry out a new test for this hypothesis through a key prediction of the Matthew effect: first mover advantage [1, 2]. Hence from the “doubling-down on intermittent success” hypothesis, one might expect that on average individual performance would decrease over time as one repeats the same recipe for success. Anecdotally, this prediction seems consistent with several examples, including those quoted by the Reviewer. For example, in terms of IMDB ratings, the first LOTR movie has a higher rating than the second one which has a higher rating than the third one. More systematically, research shows that while sequel films have good box-office performance in general, it is rare for sequels to outperform the predecessors in terms of either gross box office or the probability to earn award recognitions [3, 4]. Similarly, in science, the first study that opens up a new line of inquiry tends to be highly cited. And follow-up studies that continue the investigations may also attract attention, but are less likely to be cited at the same level as the canonical piece. Admittedly, individual cases may vary, and there are always exceptions where subsequent works overtake their forebear in prominence. But on average, the “doubling-down on intermittent success” hypothesis would predict that works produced in the early part of the hot streak should have a higher impact than those produced later in the hot streak.

Next, we perform two different measurements to test this prediction: (1) We compare the impact distribution of works produced in the first and second half of the hot streak, as measured by the logarithmic of auction price, the IMDB rating and the logarithmic of paper citations in 10 years C_{10} (Fig. R13a-c). (2) We measure the relative position of the highest-impact work among the top six highest-impact works in a career (Fig. R13d-f). Interestingly—and contrary to what the hypothesis would predict—we find no systematic difference between the impact in the first and second half of the hot streak. Moreover, we observe an equal probability for the highest-impact work to appear before and after other hits in a career. In other words, across both measurements, our results show that the impact during a hot streak stays remarkably stable, and does not exhibit decreasing trends. Overall, these results appear in tension with the “doubling-down on intermittent success” hypothesis. At the same time, these findings improve our understanding of the hot streak dynamics, further enriching the interestingness of the underlying phenomena. Indeed, these results suggest that, although individuals become substantially more focused during hot streak, they tend to work on similar but not identical topics or styles; this in part suggests that people who simply reproduce their past success may not sustain their high-level performance.

Figure R13 (a-c) The impact distribution for the first and the second half of a hot streak across three domains. (d-f) The distribution of the relative position $P(\tilde{N})$ of the three highest-impact works among the six highest-impact works within a career for artists, where \tilde{N} denotes the relative order among the top six hits.

In summary, we thank the Reviewer for this insightful comment, reflecting the Reviewer’s deep understanding of the underlying processes we aim to study. As we show here, although the King Midas golden touch and the doubling-down on intermittent success hypothesis each appears plausible, neither of them alone can fully explain the intriguing patterns observed in real careers. The additional analyses presented here not only support the novelty of our results, but also furthered our own understanding of the underlying phenomenon. It is also crucial to note that our findings on the exploration, exploitation and career hot streaks hold the same, regardless the underlying processes. Motivated by the Reviewer’s comment, we added a new section in SI S6.7 to discuss the ‘doubling-down on intermittent success’ hypotheses in more detail.

5.11 D) *Given the authors’ definitions of exploration and exploitation based upon extreme z-scores, the statements “We find that when exploitation occurs by itself” and “following an exploration episode alone” are unclear, since an individual cannot be in a hybrid exploration and exploitation state.*

Response: Thank you for this very helpful comment. It is also related to ‘*The authors seem to take it as a given, and so the rhetorical statement “Are career hot streaks reflective of exploration or exploitation behavior, or some combination of the two?” seems self-fulfilling.*’ in 5.8. We address these points together below in detail.

Here the Reviewer raised a very important point that we have not considered as carefully in our original paper. We are very grateful for this insight that the Reviewer

shared with us. Indeed, there are in principle three states: exploration, exploitation, and normal state. In the original paper, we mainly focused on the first two states, and their combinations. But the Reviewer’s comment made us realize that we could also incorporate the normal state into our framework, and further stress test our findings. Indeed, one advantage of the z-score analysis is that it not only allows us to identify exploration and exploitation phases, but also the normal state, where the topic/style diversity is not significantly different from one’s typical level. In this revision, we probe this issue through two levels of analyses. First, we show evidence for the existence of the normal phase. Second, we explicitly incorporate the normal phase into our framework of exploration and exploitation, testing all nine difference combinations of the three phases.

To establish an intuition for the normal phase, we measure the topic entropy after the end of a hot streak (Fig. R14). We find a lack of significant difference in entropy between real careers and the null model after a hot streak ends, suggesting an individual goes back to “normal” following the hot-streak period. We feel this new analysis not only helps to orient the readers to better understand the role of exploration and exploitation in the context of one’s typical behavior, it also offers new evidence for what happens after the hot streak ends. We now add this result as a new figure in the main text (Fig. 5), allowing us to expand our discussions on the end of the hot streak as well.

Figure R14. The real careers do not show significant difference to the null model after the end of a hot streak, suggesting the existence of the normal phase.

In our second analysis, we explicitly incorporate the normal phase into our framework, test all nine combinations of the three phases and correlate them with the onset of career hot streaks. To do so, we repeat the null model used for each career and identify phases of exploration, normal, and exploitation within each career by comparing the entropy in a period to the distribution $P(H)$ predicted by the null

model (Fig. R15). We define exploration, exploitation, and normal phase as a period with H significantly larger, smaller, or similar to one's typical level. We measure the probability to observe the onset of hot streak for the nine different combinations between exploration, normal phase and exploitation, and compare it to a baseline when the hot streak randomly appears in a career (Fig. R16). We find that the percentage change for the exploration-exploitation transition is significantly larger than zero and the highest among all types of combinations. Together, the additional analyses further clarify the existence of normal phases of production and the possibility of alternative strategies.

Based on the Reviewer's comments, we have now added a new section in SI S5.1 to discuss the normal phase in detail. We have also made clarifications on this point in the main text.

Figure R15 Illustration on how to define normal, exploration and exploitation phase. For any period of the same length in real careers, we compare its entropy to the null model distribution, and assign it as exploitation if the entropy is below the 25th quantile of the null model, normal phase if the entropy is between the 25th and 75th quantile, and exploration if the entropy is above the 75th quantile.

Figure R16 (a-c) Comparing the probability to observe the onset of hot streak for different combinations of exploration, normal and exploitation phases to the baseline. Color denotes the percentage change, where blue is below the baseline and red is above the baseline. Dashed box indicates the percentage change is not significantly different zero (chi-square test, $p\text{-value} \geq 0.05$) (d-f) The total percentage change of exploration, normal and exploitation phases before and during the hot streaks. The value of the bar plot is the sum of the matrix in each row (before) and each column (during).

5.12 E) The analysis regarding team size suffers from endogeneity issues commonly attributable to multiple explanatory variables - does the transition into the exploration period produce more diverse work that is more highly cited for this reason, or does the shift to larger teams produce more diverse work that is more highly cited for this different reason, not least of which is there are more coauthors to show the work.

Response: Here the Reviewer raises another important point: is the work produced during a hot streak highly cited due to the larger team size itself? One possibility, as the Reviewer pointed out, is that there are more coauthors to show the work. Indeed, this is an important point in the team science literature, and researchers have argued for a need to adjust for self-citations to account for the increased visibility from coauthors [18]. Following the literature, here we repeat our analysis by excluding self-citations. Specifically, for each citation of a paper, we compare the coauthors' last name and first initial. If they share at least one author with the same name, we consider it as a self-citation and subtract it from C_{10} , which offers a conservative estimation on the effect of self-citations. We find that although adjusted C_{10} (without self-citation) is smaller than the raw C_{10} by construction (Fig. R17a), the two values are highly correlated. We further quantify the distribution of logarithmic of adjusted C_{10} for papers published during hot streaks and normal phases for scientists in our dataset (Fig. R17b), and find that the hot streak papers still have systematically higher impact, indicating that the highly cited papers are not just due to self citations. We added a new section in SI S4.4 to discuss this new analysis in detail. We also recognize that given the descriptive nature of our analysis, it by definition cannot fully resolve the endogeneity issues, as is true for all other descriptive analyses. Here to make this point truly transparent to our readers, we also added detailed discussions in the main text to highlight this issue specifically in the paper.

Figure R17 (a) Scatter plot between raw and self-citation adjusted C_{10} for 10000 random papers in the dataset. (b) The distribution of logarithmic self-citation adjusted C_{10} for papers published during hot streaks and normal phases.

*

We would like to thank the Reviewer for these extremely insightful comments and guidance. They highlight the deep knowledge and expertise of the Reviewer, and we have benefited tremendously from his/her insights. As a result, the paper has substantially improved in its scope and richness, with **four** additional discussions in the main text, discussing domain-specific differences, data coverage, individual vs external factors, and effect size. In the Supplementary Information, we have added **eight** new sections (S1.4, S1.6, S3.11, S3.13, S4.4, S4.5, S5.1, S6.7) and revised **two** previous sections (S1.2 and S2.2), with the changes totaling **sixteen** pages. These changes also resulted in **nine** new Supplementary figures (Fig. S29-S31, S33, S42-S43, S45-S46, S48).

Overall, we feel that the manuscript has improved significantly by incorporating these insightful comments. In particular, we are encouraged by the robustness of our results against a wide range of variations. We are also very excited about the paper's potential to stimulate further work and hope that the Reviewer feels the same. We hope the Reviewer finds the paper now suitable for publication. Moreover, please don't hesitate to let us know if there is anything else we can do to further improve the piece!

Reference

1. Merton, R.K., *Matthew effect in science*. Science, 1968. **159**(3810): p. 56-&.
2. Barabási, A.-L. and R. Albert, *Emergence of scaling in random networks*. science, 1999. **286**(5439): p. 509-512.
3. Simonton, D.K., *Cinematic success criteria and their predictors: The art and business of the film industry*. Psychology & marketing, 2009. **26**(5): p. 400-420.
4. Cattani, G. and S. Ferriani, *A core/periphery perspective on individual creative performance: Social networks and cinematic achievements in the Hollywood film industry*. Organization science, 2008. **19**(6): p. 824-844.
5. Luger, J., S. Raisch, and M. Schimmer, *Dynamic balancing of exploration and exploitation: The contingent benefits of ambidexterity*. Organization Science, 2018. **29**(3): p. 449-470.
6. Burt, R.S. and J. Merluzzi, *Network oscillation*. Academy of Management Discoveries, 2016. **2**(4): p. 368-391.
7. Wu, L., D. Wang, and J.A. Evans, *Large teams develop and small teams disrupt science and technology*. Nature, 2019. **566**(7744): p. 378-382.
8. Zeng, A., et al., *Fresh teams are associated with original and multidisciplinary research*. Nature Human Behaviour, 2021: p. 1-9.
9. Zeng, A., et al., *Increasing trend of scientists to switch between topics*. Nature Communications, 2019. **10**.
10. Foster, J.G., A. Rzhetsky, and J.A. Evans, *Tradition and Innovation in Scientists' Research Strategies*. American Sociological Review, 2015. **80**(5): p. 875-908.
11. Kuhn, T.S., *The Essential Tension: Selected Studies in Scientific Tradition and Change*. 1977: University of Chicago Press.
12. March, J.G., *Exploration and exploitation in organizational learning*. Organization Science, 1991. **2**(1): p. 71-87.
13. Lavie, D., U. Stettner, and M.L. Tushman, *Exploration and Exploitation Within and Across Organizations*. Academy of Management Annals, 2010. **4**: p. 109-155.
14. Liu, L., et al., *Hot streaks in artistic, cultural, and scientific careers*. Nature, 2018. **559**(7714): p. 396.
15. Baliotti, S., R.L. Goldstone, and D. Helbing, *Peer review and competition in the Art Exhibition Game*. Proceedings of the National Academy of Sciences, 2016. **113**(30): p. 8414-8419.
16. Lazer, D. and A. Friedman, *The network structure of exploration and exploitation*. Administrative science quarterly, 2007. **52**(4): p. 667-694.
17. Liu, L., et al., *Hot streaks in artistic, cultural, and scientific careers*. Nature, 2018. **559**(7714): p. 396-+.
18. Valderas, J.M., *Why do team-authored papers get cited more?* Science, 2007. **317**(5844): p. 1496-1498.

Reviewers' Comments:

Reviewer #3:

Remarks to the Author:

I congratulate the authors for an exceptionally thorough respect to the reviewers, which addresses my concerns.

Reviewer #4:

Remarks to the Author:

The authors have made considerable revisions to their paper and in general, I am satisfied with their responses and the paper could be accepted for publication after considering the following comments.

I have only a couple of minor comments and questions.

Regarding 4.3, I realize that the authors wanted to normalize by the number of papers and indeed, this is likely necessary. For presenting results though the concept of 'topics per paper' is a bit difficult to comprehend. The inverse, "papers per topic" seems to be more comprehensible though it is not as good at highlighting the point that scientists work on fewer topics. In any case, I would encourage the authors to consider the tradeoffs of these two different approaches carefully in deciding which one is better to present.

In the caption to Fig. R5 in the response letter, it appears the authors mistakenly wrote $P(\langle n/m \rangle)$ instead of m/n as in other places, but I don't see that this error appears in the SI or main text anywhere, so I don't believe this is an issue.

A further possible concern is that the error bars in Fig. 4a appear surprisingly small. Perhaps the dataset is large enough that they have excellent statistics, but given that the number of authors on papers can vary dramatically (even up to hundreds of authors for some papers), I am surprised that the error bars on the team size are less than 0.1 authors per paper. Perhaps with a large number of authors the outliers and statistics end up working out in this way, but I would encourage the authors to just double-check these error bars.

Reviewer #5:

Remarks to the Author:

The authors have carefully addressed my initial set of comments in a satisfactory manner.

Point-By-Point Response

Reviewer #3:

3.1 *I congratulate the authors for an exceptionally thorough respect to the reviewers, which addresses my concerns.*

Response: Thank you again for all your insightful comments in the previous round that substantially helped us shape up the paper!

Reviewer #4

4.1 *The authors have made considerable revisions to their paper and in general, I am satisfied with their responses and the paper could be accepted for publication after considering the following comments.*

I have only a couple of minor comments and questions.

Response: We are delighted to hear your recommendation for publication. Next, we offer a detailed point-by-point response to your additional suggestions.

4.2 *Regarding 4.3, I realize that the authors wanted to normalize by the number of papers and indeed, this is likely necessary. For presenting results though the concept of ‘topics per paper’ is a bit difficult to comprehend. The inverse, “papers per topic” seems to be more comprehensible though it is not as good at highlighting the point that scientists work on fewer topics. In any case, I would encourage the authors to consider the tradeoffs of these two different approaches carefully in deciding which one is better to present.*

Response: We thank the Reviewer for pointing this out. Here the Reviewer suggested us to consider two measures: ‘topics per paper’ and ‘papers per topic’. The first measure gauges the number of topics given the productivity level. The second approach captures the average productivity per topic. We agree with Reviewer that the first one seems better at capturing whether individuals work on fewer/more topics within a time period. Following the Reviewer’s comment, we replaced the term ‘topics per paper’ with “the number of topics normalized the number of papers” to add further clarification for this measure.

4.3 *In the caption to Fig. R5 in the response letter, it appears the authors mistakenly wrote $P(<n/m>)$ instead of m/n as in other places, but I don't see that this error appears in the SI or main text anywhere, so I don't believe this is an issue.*

Response: We thank the Reviewer for this comment. We have double checked the figure and relevant sentences to ensure accuracy.

4.4 *A further possible concern is that the error bars in Fig. 4a appear surprisingly small. Perhaps the dataset is large enough that they have excellent statistics, but given that the number of authors on papers can vary dramatically (even up to hundreds of authors for some papers), I am surprised that the error bars on the team size are less than 0.1 authors per paper. Perhaps with a large number of authors the outliers and statistics end up working out in this way, but I would encourage the authors to just double-check these error bars.*

Response: Thank you so much for this comment. We double-checked the error bars in Fig. 4a. The error bars represent the standard error of the mean, which becomes smaller with larger sample size. We have now clarified the definition of the error bar in the figure caption.

Reviewer #5

5.1 *The authors have carefully addressed my initial set of comments in a satisfactory manner.*

Response: Thank you again for your insightful comments in the previous round that substantially helped to improve our paper.